# MUSA-PINN: Multi-scale Weak-form Physics-Informed Neural Networks for Fluid Flow in Complex Geometries

Weizheng Zhang [* 1]   Xunjie Xie [* 1]   Hao Pan [2]   Xiaowei Duan [1]   Bingteng Sun [3]   Qiang Du [3]   Lin Lu [1]

## Abstract

While Physics-Informed Neural Networks (PINNs) offer a mesh-free approach to solving fluid-flow PDEs, standard point-wise residual minimization suffers from convergence pathologies in topologically complex domains like Triply Periodic Minimal Surfaces (TPMS). The locality bias of point-wise constraints fails to propagate global information through tortuous channels, causing unstable gradients and conservation violations. To address this, we propose the Multi-scale Weak-form PINN (MUSA-PINN), which reformulates Navier-Stokes equation constraints as integral conservation laws over hierarchical spherical control volumes. We enforce continuity and momentum conservation via flux-balance residuals on control surfaces. Our method utilizes a three-scale subdomain strategy-comprising large volumes for long-range coupling, skeleton-aware meso-scale volumes aligned with transport pathways, and small volumes for local refinement-alongside a two-stage training schedule prioritizing continuity. Experiments on steady incompressible flow in TPMS geometries show MUSA-PINN outperforms state-of-the-art baselines, reducing relative errors by up to 93% and preserving mass conservation.

## 1. Introduction

Scientific machine learning has advanced rapidly in recent years, offering new paradigms for modeling and solving complex physical processes. Yet in engineering fluid dynamics, conventional computational fluid dynamics (CFD) remains the dominant tool (Slotnick et al., 2014; Versteeg, 2007), with costs that become prohibitive in parameterized design and optimization loops. High-efficiency heat exchangers exemplify this challenge: their performance relies on geometrically intricate three-dimensional flow passages, and practical design exploration often requires repeated simulations across many geometric variations. Such simulations via CFD typically require high-quality volumetric meshes, but mesh generation for slender, tortuous, and topologically complex channels is time-consuming and frequently demands substantial expert effort and manual tuning (Hughes et al., 2005).

Physics-informed neural networks (PINNs) provide a mesh-free alternative by representing the solution field with neural networks and training with physics constraints from governing equations and boundary conditions (Raissi et al., 2019). PINNs are attractive for fast inference, differentiable modeling, and inverse problems with sparse observations (Karniadakis et al., 2021; Raissi et al., 2020). For high-dimensional fluid simulations, PINNs have the potential to significantly improve computational efficiency, in contrast to traditional CFD methods whose computational cost increases dramatically with problem dimensionality (Hwang, 2025). However, applying PINNs to industrial-grade internal flows with tortuous, high-aspect-ratio, and topologically complex channels remains challenging. Empirically, the accuracy and stability of standard PINNs often degrade substantially as geometric complexity increases, even for steady incompressible flows. Figure 1 provides a qualitative comparison of flow streamlines on an industrial liquid-cooling plate, contrasting CFD with a standard PINN baseline and MUSA-PINN. In this paper, we make the observation that a key reason for this failure is structural: the dominant training signal in standard PINNs is pointwise residual minimization, which cannot match with the global nature of conservation laws and produces a local–global inconsistency in complex transport geometries. In long and winding passages as found in heat exchangers, where information propagation along streamlines and across junctions is weak under purely local supervision, such locally small residuals frequently translate into physically inconsistent behavior.

A natural remedy to the above problem is to impose conservation in an integral sense. Prior attempts include enforcing

---

[1]Shandong University, Qingdao, China [2]Tsinghua University, Beijing, China [3]Institute of Engineering Thermophysics, Chinese Academy of Sciences, Beijing, China. Correspondence to: Lin Lu <llu@sdu.edu.cn>.

*Proceedings of the 43rd International Conference on Machine Learning*, Seoul, South Korea. PMLR 306, 2026. Copyright 2026 by the author(s).

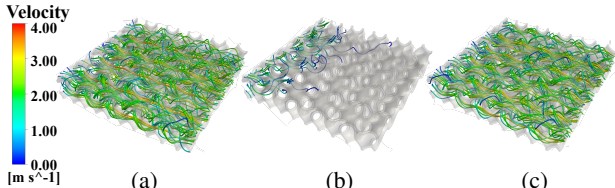

*Figure 1.* **Flow streamlines on an industrial liquid-cooling plate.** (a) CFD reference, (b) PINN-PE baseline (degrades in this complex geometry), and (c) MUSA-PINN. MUSA-PINN achieves a $14.91\%$ relative $\ell_2$ error in velocity w.r.t. CFD.

global flux constraints between inlets and outlets (Hwang, 2025) or adopting variational formulations with element-wise quadrature (Kharazmi et al., 2021). Yet, global boundary constraints are sparse and may leave the interior under-constrained, while element-based variational formulations typically require meshing and may introduce additional computational burdens or stability issues on complex geometries. This motivates a mesh-free formulation that enforces conservation over interior regions, so that physical information can propagate reliably through tortuous channels instead of being constrained only at scattered points or the outer boundary.

In this work, we propose MUlti-Scale weAk-form PINN (MUSA-PINN), a hybrid strong–weak form physics-informed framework for steady incompressible flow in complex geometries. Inspired by the classical Finite Element Analysis using weak form integral equations to improve convergence, our main idea is to reformulate governing equations into flux-balance constraints over control volumes using the Gauss divergence theorem, turning volumetric residuals into surface integrals of physically meaningful fluxes. These surface integrals are efficiently approximated by Monte Carlo sampling on control-volume boundaries, avoiding volumetric meshing while providing a more structured and globally informative training signal than pointwise residuals. To make the weak-form constraints effective across different industrial channels, we further introduce a multi-level subdomain placement strategy: large-scale control volumes encourage long-range conservation across transport pathways; medium-scale control volumes trace a geometry skeleton, so conservation signals follow flow channels; and small-scale control volumes refine local details and stabilize near-boundary behaviors. Together, these components provide dense conservation supervision throughout the domain and mitigate the local–global mismatch that limits conventional PINNs in complex internal flows.

We evaluate MUSA-PINN on flow channels extracted from Triply Periodic Minimal Surface (TPMS) structures, which serve as representative heat-exchanger passages due to their complex geometry and favorable transfer characteristics (Yeranee & Rao, 2022). Experiments on multiple TPMS

variants demonstrate that MUSA-PINN significantly improves solution accuracy and physical consistency over state-of-the-art baselines, and ablations verify the contributions of the weak-form and multi-scale designs. We further discuss extensions toward more complex industrial components (e.g., liquid-cooling plates and DualMS channels (Zhang et al., 2025)) to highlight the scalability of the proposed approach.

Our main contributions are summarized as follows:

- Weak-form conservation via control volumes. We introduce a divergence-theorem-based formulation that enforces mass and momentum conservation through surface flux integrals over spherical control volumes, providing a volumetric mesh free and physically structured supervision signal beyond pointwise residuals.
- Multi-scale subdomain placement for complex channels. We design a multi-level strategy that improves coverage and information propagation in long, tortuous, and branching internal passages, bridging global consistency and local fidelity without manual domain decomposition.
- We benchmark on geometrically intricate TPMS heat-exchanger channels and conduct systematic comparisons and ablations, demonstrating consistent gains in accuracy and recovered flow structures, and outlining scalability to more complex industrial geometries.

## 2. Related Work

PINNs solve PDEs by parameterizing solution fields with neural networks and enforcing governing equations and boundary/initial conditions via automatic differentiation (Raissi et al., 2019; Cuomo et al., 2022). Recent studies have improved and analyzed PINNs training, including advances in architectures, optimization strategies, sampling schemes, and theoretical perspectives on when and why PINNs succeed or fail (Karniadakis et al., 2021; de la Mata et al., 2023; Toscano et al., 2025; Krishnapriyan et al., 2021). These developments position PINNs as a flexible learning-based solver family that can be deployed across diverse physical systems (Cai et al., 2021; Huang & Wang, 2022; Wang et al., 2023; Hu et al., 2024).

**PINNs for Navier-Stokes equations.** PINNs have been applied to incompressible Navier–Stokes flows under various settings, including data-scarce or label-free formulations and enhanced representations (Jin et al., 2021; Chen et al., 2021; Bai et al., 2020; Gao et al., 2021). While substantial progress has been demonstrated on many 2D benchmarks and simple 3D domains, steady flow prediction in 3D complex channels remains challenging due to intricate geometry, long-range coupling, and optimization instability. We target this regime and propose a weak-form, multi-scale formula-

tion tailored to complex 3D geometries.

**Strong-form PINNs and Weak-form PINNs.** Strong-form PINNs minimize pointwise PDE residuals, offering a clean mesh-free pipeline but often exhibiting unstable optimization and weak global coupling in complex geometries. To mitigate these issues, many improvements have been proposed around adaptive sampling strategies (Lu et al., 2021; Zapf et al., 2022; Wu et al., 2023), dynamic loss reweighting (Wang et al., 2021a; 2022; Maddu et al., 2022), and training strategies (D. Jagtap & Em Karniadakis, 2020; Jagtap et al., 2020; Wang et al., 2024; 2025b), aiming to balance competing residual terms and stabilize convergence. Region-based approaches like RoPINN (Wu et al., 2024) mitigate local noise by averaging residuals over continuous neighborhoods, but remain strong-form that requires higher order differentiation. Furthermore, they rely on random sampling that ignores topological structures of local neighborhoods. Weak-form and variational neural solvers provide an alternative by enforcing integrated residuals or variational principles. Representative examples include the Deep Ritz method (Yu et al., 2018), weak adversarial networks (Zang et al., 2020), and variational PINNs such as V-PINN (Kharazmi et al., 2019) and hp-VPINN (Kharazmi et al., 2021). These methods reduce derivative order or improve stability, but often rely on predefined elements, test-function bases, or domain decomposition. In contrast, MUSA-PINN applies the divergence theorem on clipped spherical control volumes to convert volume-integrated conservation laws into Hessian-free surface flux balances over interior control surfaces and clipped physical boundaries, avoiding volumetric meshing and volumetric quadrature.

**Multi-scale PINNs.** Existing multi-scale PINNs mainly address spectral bias or stiff coupling and can be grouped into three paradigms. Frequency-domain architectures combat spectral bias: MscaleDNN (Liu et al., 2020) employs frequency-scaled subnetworks to decompose target functions, while Fourier feature networks (Wang et al., 2021b) map coordinates into a Fourier embedding space. Optimization-centric strategies (e.g., Multi-Adam (Yao et al., 2023)) rebalance gradients across stiff loss terms via scale-invariant updates. Physical domain decomposition leverages physical priors to separate disparate spatio-temporal scales through reaction-rate grouping and boundary-layer/multiphysics decompositions (Weng & Zhou, 2022; Huang et al., 2024; Chaffart et al., 2024). However, they do not explicitly tackle the geometric spatial multi-scales in industrial complex domains, where flow hierarchy is governed by channel connectivity rather than frequency content or idealized priors. Our method targets this setting by enforcing integral conservation over multi-scale control volumes and placing them in a topology-aware fashion to propagate constraints along complex channels.

# 3. Methodology

We propose MUSA-PINN, a hybrid strong–weak physics-informed framework for steady incompressible flow in complex geometries (Figure 2). Starting from a standard strong-form PINN that minimizes pointwise Navier–Stokes residuals and boundary-condition violations, we additionally impose weak-form conservation constraints over local integration subdomains. By integrating the governing equations over these subdomains and applying the divergence theorem, the weak constraints reduce to boundary flux-balance residuals that can be evaluated through surface integrals. To make the weak constraints effective and efficient on industrial-scale complex geometries, we further introduce a three-level multi-scale strategy to place integration subdomains at global, skeleton-aware, and local interior locations. This section first present the governing equations, then derives weak-form residuals, introduces the multi-scale subdomain construction, and finally presents the overall optimization objective.

### 3.1. Preliminaries

**Problem Setting.** We consider steady incompressible viscous flow in a complex three-dimensional geometry, a regime widely encountered in industrial internal-flow components (e.g., ducts and heat-exchanger passages) where the working fluid can often be treated as incompressible and a steady-state approximation is appropriate. Let $\Omega \subset \mathbb{R}^3$ denote the fluid domain and $\partial\Omega$ its boundary. We decompose the boundary into inlet, outlet, and wall parts:

$$\partial\Omega = \Gamma_{\text{in}} \cup \Gamma_{\text{out}} \cup \Gamma_{\text{w}}. \tag{1}$$

**Nondimensional steady Navier–Stokes.** We solve the nondimensional steady incompressible Navier–Stokes equations:

$$\begin{aligned} \nabla \cdot \mathbf{u} &= 0, && \text{in } \Omega, \\ (\mathbf{u} \cdot \nabla)\mathbf{u} + \nabla p - \frac{1}{Re}\nabla^2\mathbf{u} &= \mathbf{0}, && \text{in } \Omega, \end{aligned} \tag{2}$$

where $\mathbf{u}(\mathbf{x}) \in \mathbb{R}^3$ and $p(\mathbf{x}) \in \mathbb{R}$ denote the velocity and pressure fields, respectively, and $Re$ is the Reynolds number. The Reynolds number definition is given in Appendix A.1.

**Boundary conditions.** We enforce standard boundary conditions for internal flows: prescribed inlet velocity $\mathbf{u} = \mathbf{u}_{\text{in}}(\mathbf{x})$ on $\Gamma_{\text{in}}$, prescribed outlet pressure $p = p_{\text{out}}(\mathbf{x})$ on $\Gamma_{\text{out}}$, and no-slip walls $\mathbf{u} = \mathbf{0}$ on $\Gamma_{\text{w}}$.

**PINN parameterization.** We approximate the solution by a neural network $f_\theta : \mathbb{R}^3 \to \mathbb{R}^4$ that outputs

$$f_\theta(\mathbf{x}) = \big(\mathbf{u}_\theta(\mathbf{x}), \, p_\theta(\mathbf{x})\big), \tag{3}$$

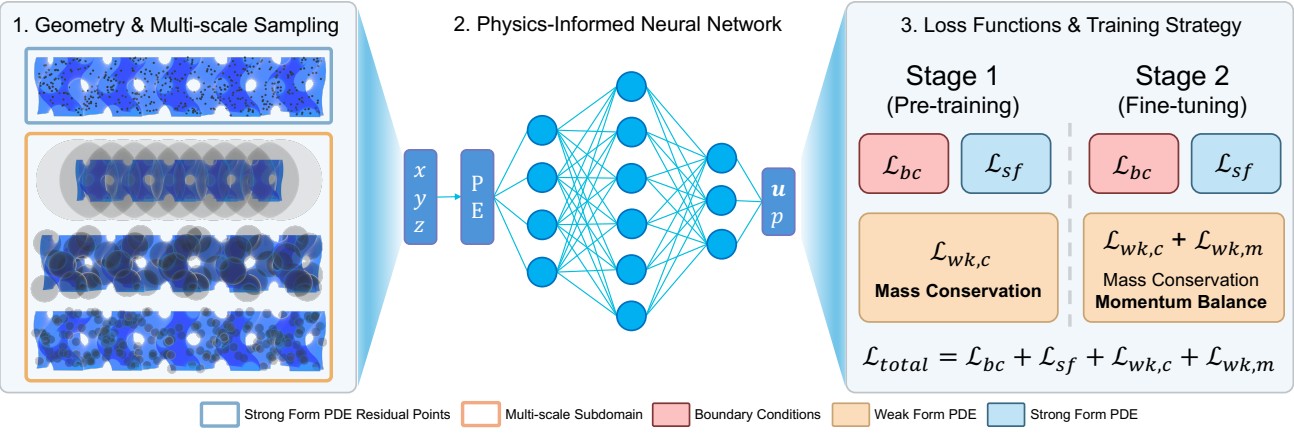

*Figure 2.* **MUSA-PINN pipeline.** We sample interior and boundary points for strong-form training and place multi-scale control volumes to impose weak conservation via surface fluxes. A coordinate-to-field MLP with positional encoding maps $(x, y, z) \mapsto (\mathbf{u}, p)$ and is trained in two stages: Stage 1 uses $\mathcal{L}_{bc} + \mathcal{L}_{sf} + \mathcal{L}_{wk,c}$, and Stage 2 adds $\mathcal{L}_{wk,m}$, forming $\mathcal{L}_{total}$.

where $\mathbf{u}_\theta(\mathbf{x}) \in \mathbb{R}^3$ and $p_\theta(\mathbf{x}) \in \mathbb{R}$ denote the predicted velocity and pressure fields, respectively. Let $\mathcal{X}_{\mathrm{sf}} \subset \Omega$ be interior collocation points for enforcing the strong-form equations, and $\mathcal{X}_{\mathrm{bc}} \subset \partial\Omega$ be boundary points for enforcing boundary conditions. The strong-form residuals at $\mathbf{x} \in \mathcal{X}_{\mathrm{sf}}$ are defined as

$$r_c(\mathbf{x}) := \nabla \cdot \mathbf{u}_\theta(\mathbf{x}),$$

$$\mathbf{r}_m(\mathbf{x}) := (\mathbf{u}_\theta(\mathbf{x}) \cdot \nabla)\mathbf{u}_\theta(\mathbf{x}) + \nabla p_\theta(\mathbf{x}) - \frac{1}{Re}\nabla^2\mathbf{u}_\theta(\mathbf{x}).$$
$$(4)$$

These strong-form terms provide pointwise physics supervision and serve as baseline constraints in our hybrid objective.

### 3.2. Weak Formulation via Divergence Theorem

While the strong-form residuals enforce the PDE pointwise, they can be challenging to optimize on complex geometries. We therefore augment the strong-form PINN with weak-form conservation constraints defined over local integration subdomains. Importantly, our weak-form constraints are derived from the same strong-form Navier–Stokes equations and are imposed as additional supervision signals, rather than replacing the strong-form terms.

**Local integration subdomains.** For a center $\mathbf{c} \in \Omega$ and a radius $r > 0$, we define a spherical subdomain intersected with the fluid region:

$$V(\mathbf{c}, r) := B(\mathbf{c}, r) \cap \Omega, \quad (5)$$

where $B(\mathbf{c}, r) = \{\mathbf{x} \in \mathbb{R}^3 : \|\mathbf{x} - \mathbf{c}\| \leq r\}$. As illustrated in Figure 3, $\partial V(\mathbf{c}, r)$ decomposes into the spherical part and the clipped geometry part:

$$\partial V(\mathbf{c}, r) = \big(\partial B(\mathbf{c}, r) \cap \Omega\big) \cup \big(B(\mathbf{c}, r) \cap \partial\Omega\big), \quad (6)$$

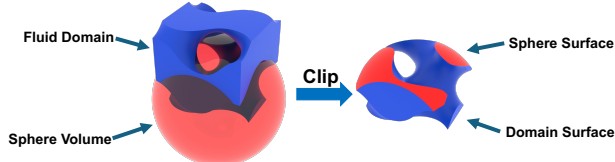

*Figure 3.* **Clipped integration subdomain.** A sphere $B(\mathbf{c}, r)$ (red) is intersected with the fluid domain $\Omega$ (blue) to form $V(\mathbf{c}, r)$.

corresponding to the spherical part and the clipped boundary part. Implementation details for constructing $V(\mathbf{c}, r)$ are provided in Appendix A.2.

**Weak-form continuity constraint.** Starting from $\nabla \cdot \mathbf{u} = 0$, integrating over $V(\mathbf{c}, r)$ and applying the divergence theorem yield

$$\int_{V(\mathbf{c},r)} \nabla \cdot \mathbf{u}\,dV = \oint_{\partial V(\mathbf{c},r)} \mathbf{u} \cdot \mathbf{n}\,dS = 0, \quad (7)$$

where $\mathbf{n}$ denotes the outward unit normal on $\partial V(\mathbf{c}, r)$.

We define the weak-form continuity residual for the network prediction as

$$R_c(\mathbf{c}, r) := \oint_{\partial V(\mathbf{c},r)} \mathbf{u}_\theta \cdot \mathbf{n}\,dS. \quad (8)$$

**Weak-form momentum constraint.** We rewrite the steady momentum equation in a divergence form. Using incompressibility, one practical nondimensional form is

$$\nabla \cdot (\mathbf{u} \otimes \mathbf{u}) + \nabla p - \frac{1}{Re}\nabla^2\mathbf{u} = \mathbf{0}. \quad (9)$$

Noting that $\nabla^2\mathbf{u} = \nabla \cdot (\nabla\mathbf{u})$, we obtain

$$\nabla \cdot \left(\mathbf{u} \otimes \mathbf{u} + p\mathbf{I} - \frac{1}{Re}\nabla\mathbf{u}\right) = \mathbf{0}. \quad (10)$$

Integrating over $V(\mathbf{c}, r)$ and applying the divergence theorem yield the boundary flux-balance condition

$$\oint_{\partial V(\mathbf{c},r)} \left( \mathbf{u} \otimes \mathbf{u} + p\mathbf{I} - \frac{1}{Re}\nabla\mathbf{u} \right) \mathbf{n}\, dS = \mathbf{0}. \quad (11)$$

Accordingly, we define the weak-form momentum residual as

$$\mathbf{R}_m(\mathbf{c}, r) := \oint_{\partial V(\mathbf{c},r)} \left( \mathbf{u}_\theta \otimes \mathbf{u}_\theta + p_\theta\mathbf{I} - \frac{1}{Re}\nabla\mathbf{u}_\theta \right) \mathbf{n}\, dS. \quad (12)$$

In practice, the surface integrals in Eq. (8) and Eq. (12) are evaluated by a volumetric mesh free Monte Carlo surface sampling scheme; details are provided in Appendix A.3.

### 3.3. Multi-scale Subdomain Placement Strategy

Industrial internal-flow geometries often exhibit challenging multi-scale characteristics, including long-range transport along slender passages, branching structures, and fine-scale geometric details. In such settings, weak-form constraints defined on a single scale can suffer from insufficient coverage or biased supervision: large subdomains emphasize global consistency but may miss local details, whereas small subdomains provide local refinement but may fail to propagate long-range conservation across the domain. To make weak-form enforcement both effective and efficient, we introduce a three-level multi-scale strategy that places integration subdomains at complementary locations and scales.

We consider three radii $r_L > r_M > r_S$ and construct three sets of subdomain centers, $\mathcal{C}_L$, $\mathcal{C}_M$, and $\mathcal{C}_S$. The radii are set from geometry-aware length scales, with details given in Appendix A.5. Each center $\mathbf{c} \in \mathcal{C}_s$ (with $s \in \{L, M, S\}$) defines an integration subdomain $V(\mathbf{c}, r_s) = B(\mathbf{c}, r_s) \cap \Omega$, on which the weak-form residuals in Eq. (8) and Eq. (12) are evaluated via the Monte Carlo estimator in Sec. 3.2.

**Large-scale subdomains.** We place large-radius subdomains to provide global coverage and enforce long-range conservation. Concretely, we sample centers $\mathcal{C}_L$ throughout the geometry's bounding box and keep those that lie inside $\Omega$. These large subdomains couple distant regions through flux-balance constraints, stabilizing the global flow organization and reducing large-scale drift that may arise from purely pointwise supervision.

**Medium-scale subdomains.** To better cover slender and branching passages, we place medium-radius subdomains along the flow channel's topological skeleton, as illustrated in Figure 4, which approximates the transport pathways inside $\Omega$. Specifically, we extract a set of skeleton samples and use them as centers $\mathcal{C}_M$. This skeleton-aware placement yields contiguous weak-form coverage along the dominant

flow routes and mitigates coverage holes that can occur with purely random interior sampling, especially in long tortuous channels and near junctions. We extract this skeleton using the Mean Curvature Flow algorithm (Tagliasacchi et al., 2012), which iteratively contracts the domain into a medial curve network. The resulting graph is then uniformly sampled to define the final center positions $\mathcal{C}_M$.

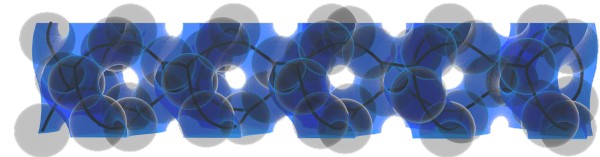

*Figure 4.* Medium-scale subdomains. These subdomains are centered along the skeletons (grey curves) of flow channels and have diameters that cover channel width.

**Small-scale subdomains.** Finally, we sample small-radius subdomains by drawing centers $\mathcal{C}_S$ uniformly inside $\Omega$. These local subdomains act as a refinement mechanism, strengthening weak-form supervision in regions with fine-scale geometric variations and helping reduce localized errors. Together with the large- and medium-scale sets, they provide dense local constraints without sacrificing global consistency.

The three scales play complementary roles: large subdomains encourage global consistency, skeleton-aware medium-scale subdomains ensure robust coverage along transport pathways, and small subdomains provide local refinement. In Sec. 3.4, we combine the weak-form losses aggregated over $\mathcal{C}_L, \mathcal{C}_M, \mathcal{C}_S$ with standard strong-form PINN losses to form the overall training objective.

### 3.4. Optimization Objective

Our training objective combines standard strong-form PINN losses with the multi-scale weak-form conservation losses. We minimize a weighted sum of interior PDE residuals, boundary-condition residuals, and weak-form flux balance residuals evaluated on the three-scale subdomain sets.

**Strong-form loss.** For interior collocation points $\mathcal{X}_{\mathrm{sf}}$, we penalize the pointwise continuity and momentum residuals:

$$\mathcal{L}_{\mathrm{sf}} = \frac{1}{|\mathcal{X}_{\mathrm{sf}}|} \sum_{\mathbf{x} \in \mathcal{X}_{\mathrm{sf}}} \left( \lambda_c |r_c(\mathbf{x})|^2 + \lambda_m \|\mathbf{r}_m(\mathbf{x})\|_2^2 \right), \quad (13)$$

where $r_c$ and $\mathbf{r}_m$ are defined in Eq. (4).

For boundary points $\mathcal{X}_{\mathrm{bc}} \subset \partial\Omega$, we enforce inlet velocity, outlet pressure, and no-slip walls via

$$\mathcal{L}_{\mathrm{bc}} = \lambda_{\mathrm{in}}\mathcal{L}_{\mathrm{in}} + \lambda_{\mathrm{out}}\mathcal{L}_{\mathrm{out}} + \lambda_{\mathrm{w}}\mathcal{L}_{\mathrm{w}}, \quad (14)$$

where $\mathcal{L}_{\mathrm{in}}$, $\mathcal{L}_{\mathrm{out}}$, and $\mathcal{L}_{\mathrm{w}}$ are mean-squared penalties on $\Gamma_{\mathrm{in}}$, $\Gamma_{\mathrm{out}}$, and $\Gamma_{\mathrm{w}}$, respectively. The full expressions and the definitions of $\mathcal{L}_{\mathrm{in}}$, $\mathcal{L}_{\mathrm{out}}$, and $\mathcal{L}_{\mathrm{w}}$ are provided in Appendix A.6.

**Multi-scale weak-form losses.** For each scale $s \in \{L, M, S\}$ with radius $r_s$ and centers $\mathcal{C}_s$, we define the continuity and momentum weak residual losses:

$$\mathcal{L}_{\mathrm{wk},c} = \sum_{s \in \{L,M,S\}} \lambda_c^s \frac{1}{|\mathcal{C}_s|} \sum_{\mathbf{c} \in \mathcal{C}_s} |R_c(\mathbf{c}, r_s)|^2, \quad (15)$$

$$\mathcal{L}_{\mathrm{wk},m} = \sum_{s \in \{L,M,S\}} \lambda_m^s \frac{1}{|\mathcal{C}_s|} \sum_{\mathbf{c} \in \mathcal{C}_s} \|\mathbf{R}_m(\mathbf{c}, r_s)\|_2^2, \quad (16)$$

where $R_c$ and $\mathbf{R}_m$ are defined in Eq. (8) and Eq. (12) and evaluated via the Monte Carlo surface estimator (Appendix A.3).

**Stage-gated overall objective.** We minimize the following training objective:

$$\min_\theta \mathcal{L}(\theta; t) = \mathcal{L}_{\mathrm{sf}} + \mathcal{L}_{\mathrm{bc}} + \mathcal{L}_{\mathrm{wk},c} + \gamma(t)\,\mathcal{L}_{\mathrm{wk},m}, \quad (17)$$

where $t$ is the training iteration and $\gamma(t)$ controls whether the weak-form momentum constraint is active. In our two-stage schedule, we set

$$\gamma(t) = \begin{cases} 0, & t < T_{\mathrm{switch}} \quad \text{(Stage I: continuity warm-up)}, \\ 1, & t \geq T_{\mathrm{switch}} \quad \text{(Stage II: enable momentum)}. \end{cases} \quad (18)$$

Stage I provides an integral incompressibility constraint that quickly suppresses long-range mass imbalance. Stage II activates the momentum flux-balance term, which further refines the flow field and improves physical consistency in complex passages. We choose $T_{\mathrm{switch}}$ as a fixed iteration or when $\mathcal{L}_{\mathrm{wk},c}$ plateaus.

**Consistency perspective.** We do not claim a formal convergence theorem for MUSA-PINN. Instead, our theoretical support is consistency-based. If $(\mathbf{u}, p)$ is a classical solution of the steady incompressible Navier–Stokes equations satisfying the boundary conditions, then the strong-form residuals vanish pointwise. Moreover, for any admissible clipped control volume $V(c, r)$ with piecewise smooth boundary, integrating the governing equations over $V(c, r)$ and applying the divergence theorem gives $R_c(c, r) = 0$ and $R_m(c, r) = 0$. Thus, the exact solution has zero population loss for both the strong-form and weak-form constraints. The Monte Carlo surface estimator used for the weak residuals is also consistent under area-uniform sampling, so the empirical weak losses converge to their population counterparts as the number of surface samples increases. This is aligned with existing PINN theory for Navier–Stokes equations, which relates approximation error to residual/training error, network size, and quadrature accuracy under suitable assumptions (De Ryck et al., 2024).

## 4. Experiments

### 4.1. Experimental Setup

**Problem Setting.** We evaluate the proposed MUSA-PINN on solving the 3D steady-state incompressible Navier-Stokes equations within four widely used high-efficiency TPMS-based heat-exchanger channels: **Primitive (P), Gyroid (G), Diamond (D), and IWP**. These structures exhibit highly tortuous, interconnected pathways that induce complex secondary flows, posing severe challenges for numerical optimization. For each geometry, we tile one TPMS unit cell in a $1 \times 1 \times 5$ arrangement along the streamwise direction and consider the bounding box $\mathcal{D} = [0, 5] \times [0, 1] \times [0, 1]$, from which the fluid region $\Omega \subset \mathcal{D}$ is extracted (details in Appendix B.1). Ground truth solutions are generated using high-fidelity CFD simulations via Ansys CFX. Detailed solver configurations are provided in Appendix B.2.

**Baselines.** We benchmark our framework against conventional PINNs and three state-of-the-art physics-informed paradigms. To ensure a rigorous evaluation, we upgrade baselines with feature embeddings to mitigate spectral bias:

(1) **PINN-PE**: Standard strong-form PINN with deterministic Fourier-feature positional encoding using linearly spaced frequencies; we found it more stable than random Fourier features (Tancik et al., 2020) in our setting (details in Appendix C).

(2) **hp-VPINN** (Kharazmi et al., 2021): a variational weak-form PINN with domain decomposition, included as a representative weak-form PINN baseline.

(3) **RoPINN** (Wu et al., 2024): A region-optimized approach that dynamically re-weights residuals based on local loss distributions.

(4) **CoPINN** (Duan et al., 2025): A cognitive learning framework employing adaptive sampling strategies.

(5) **MDPINN-GD** (Hwang, 2025): A multi-domain PINN that couples subdomains and enforces global dynamics constraints.

**Implementation Details.** We adopt a unified training protocol to enable fair comparisons. All paradigms use the same network architecture and SOAP optimizer (Wang et al., 2025a), with matched learning-rate schedules and loss weights, minimizing solver-induced confounders. Further details are given in Appendix C.

### 4.2. Comparative Analysis on Complex Geometries

**Metrics.** We report two complementary error measures for both velocity and pressure: mean squared error (MSE), and relative $\ell_2$ error over the evaluation set in Table 1.

**Performance Analysis.** Across all methods, the Primitive (P) channel is comparatively easy: all baselines achieve low

*Table 1.* **Quantitative comparison on TPMS channels.** We report **MSE** and **relative $\ell_2$ error** for velocity magnitude ($|\mathbf{u}|$) and pressure ($p$) across four geometries. **Bold** indicates the best performance, and underlined indicates the second best. **IMP.** denotes the relative improvement of **MUSA-PINN** over the best baseline for each metric.

| Method | Ref | Primitive (P) | | | | Gyroid (G) | | | | Diamond (D) | | | | IWP | | | |
|---|---|---|---|---|---|---|---|---|---|---|---|---|---|---|---|---|---|
| | | $|\mathbf{u}|$ | | $p$ | | $|\mathbf{u}|$ | | $p$ | | $|\mathbf{u}|$ | | $p$ | | $|\mathbf{u}|$ | | $p$ | |
| | | MSE | $\ell_2$ | MSE | $\ell_2$ | MSE | $\ell_2$ | MSE | $\ell_2$ | MSE | $\ell_2$ | MSE | $\ell_2$ | MSE | $\ell_2$ | MSE | $\ell_2$ |
| PINN-PE | – | 0.0006 | 3.76% | 0.0845 | 5.36% | 2.2042 | 83.19% | 105.6067 | 86.21% | 2.7627 | 90.18% | 306.4535 | 83.04% | 6.1360 | 94.49% | 754.1458 | 101.69% |
| hp-VPINN | CMAME 2021 | 0.2980 | 84.51% | 4.5096 | 39.13% | 2.8201 | 94.10% | 122.9809 | 93.06% | 3.0757 | 94.93% | 409.6463 | 95.74% | 6.4018 | 96.94% | 766.24036 | 102.50% |
| RoPINN | NeurIPS 2024 | 0.0005 | 3.51% | 0.0948 | 5.67% | 1.7492 | 74.11% | 135.9094 | 97.80% | 2.5541 | 86.71% | 360.9349 | 90.12% | 6.3375 | 96.45% | 759.8746 | 102.07% |
| CoPINN | ICML 2025 | 0.0003 | 2.83% | 0.1258 | 6.54% | 2.1992 | 83.10% | 105.5943 | 86.21% | 2.7611 | 90.15% | 306.4326 | 83.04% | 6.1125 | 94.72% | 752.9209 | 101.60% |
| MDPINN-GD | NeurIPS 2025 | 0.0002 | 2.60% | 10.6390 | 60.01% | 0.6457 | 45.03% | 14.5325 | 31.98% | 2.6693 | 88.64% | 300.3825 | 82.22% | 5.8951 | 93.02% | 686.5231 | 97.02% |
| **MUSA-PINN** | **Ours** | 0.0001 | 1.42% | 0.0052 | 1.33% | 0.0119 | 6.13% | 2.3906 | 12.97% | 0.0527 | 12.46% | 31.3286 | 26.55% | 0.0255 | 6.12% | 96.4645 | 36.37% |
| IMP. | – | 50% | 45.38% | 93.85% | 75.19% | 98.16% | 86.39% | 83.55% | 59.44% | 97.94% | 85.63% | 89.57% | 67.71% | 99.57% | 93.42% | 85.95% | 62.51% |

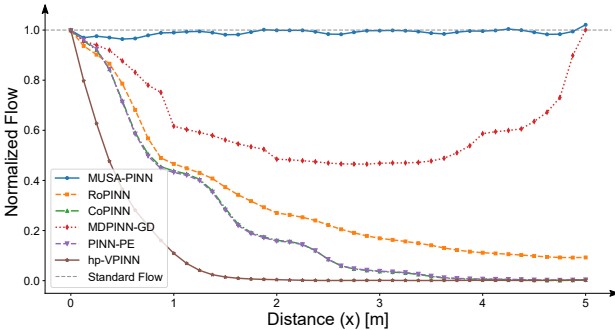

*Figure 5.* **Evaluation of Global Mass Conservation.** The plot shows the streamwise evolution of normalized mass flow rate $Q(x)/Q_{in}$ along the flow direction in the Gyroid structure. While standard methods exhibit significant mass leakage deviating from the theoretical value (gray line), MUSA-PINN maintains flux continuity, demonstrating the efficacy of volumetric constraints.

MSE and $\ell_2$ errors, indicating that standard point-wise supervision can be sufficient when the geometry is less challenging. On the Gyroid (G) channel, the optimization becomes notably harder: among the baselines, only MDPINN-GD remains stable and produces non-degenerate solutions, while other point-wise methods frequently exhibit divergence or severe accuracy degradation. On the most challenging Diamond (D) and IWP channels, the gap becomes decisive. Most baselines collapse to physically inconsistent solutions, with relative errors exceeding $90\%$ despite satisfying boundary conditions. In contrast, MUSA-PINN maintains stable convergence and achieves consistently low MSE and relative $\ell_2$ errors across all four geometries. These results suggest that enforcing integral conservation on densely placed multi-scale control volumes provides a stronger interior supervision signal, which improves robustness in tortuous, highly connected passages where point-wise constraints struggle to propagate global physical consistency.

### 4.3. Analysis of Physical Consistency

To investigate the source of MUSA-PINN's accuracy, we perform a detailed qualitative analysis focusing on global conservation and local field reconstruction.

*Table 2.* **Ablation on scale placement and weak-form schedule.** We report relative $\ell_2$ errors on velocity magnitude and pressure.

| Variant | Rel. $\ell_2$ on $|\mathbf{u}| \downarrow$ | Rel. $\ell_2$ on $p \downarrow$ |
|---|---|---|
| Large-only ($\mathcal{C}_L$) | 14.59% | 87.41% |
| Medium-only ($\mathcal{C}_M$) | 87.72% | 98.82% |
| Small-only ($\mathcal{C}_S$) | 89.99% | 98.83% |
| Continuity-only ($\mathcal{L}_{\mathrm{wk},c}$) | 13.28% | 44.19% |
| Momentum-only ($\mathcal{L}_{\mathrm{wk},m}$) | 95.87% | 99.98% |
| Joint (no staging) | 95.31% | 99.98% |
| **Full (ours, two-stage)** | **12.46%** | **26.55%** |

**Global Mass Conservation.** A critical failure mode of standard PINNs in long, winding channels is the violation of mass conservation due to error accumulation. We quantify this by measuring the mass flow rate mismatch $Q(x)/Q_{in}$ along the streamwise direction $x$ in structure Gyroid, where $Q(x)$ is the cross-sectional volumetric flow rate at $x$ and $Q_{in}$ is its inlet value. As shown in Figure 5, baselines exhibit significant mass leakage within the domain. Although MDPINN-GD explicitly enforces flux balance at boundaries, it lacks dense internal constraints. MUSA-PINN, leveraged by large-scale integral constraints, effectively acts as a volumetric regularizer, enforcing strict flux continuity throughout the entire domain. This results in a flow rate profile that tightly adheres to the conservation law, reducing the average cross-sectional mass error by over 97.05% compared to the best baseline.

**Local Feature.** Figure 6 visualizes the velocity magnitude slices in the high-curvature regions of the Gyroid structure. Baseline methods tend to produce over-smoothed solutions, failing to resolve the sharp velocity gradients near the surface boundaries. In contrast, MUSA-PINN utilizes multi-scale spherical constraints as dense, localized feature detectors. As highlighted in the zoomed-in views, our method successfully captures intricate secondary flows and boundary layer profiles that are washed out by baseline approaches, validating the capability of our weak-form formulation to preserve high-frequency physical details.

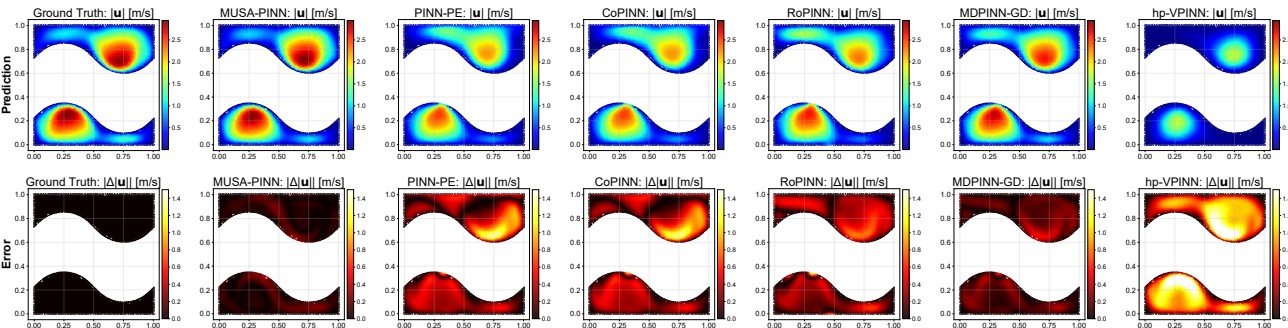

*Figure 6.* **Qualitative comparison of velocity predictions on the Gyroid slice at** $x = 0.5$. MUSA-PINN better preserves the component-wise flow structures and yields consistently lower absolute errors than the baselines. More details are provided in Appendix D.1 and D.2.

## 4.4. Ablation Studies

We ablate MUSA-PINN along two axes: *scale placement* and *weak-form design and schedule* in structure Diamond. For scale placement, we enable weak constraints on only one center set at a time: Large-only ($\mathcal{C}_L$), Medium-only ($\mathcal{C}_M$), and Small-only ($\mathcal{C}_S$). For weak-form design and schedule, we use the full multi-scale placement but vary the enforced laws and schedule: Continuity-only, Momentum-only, Joint (no staging)(continuity+momentum from the start), and Full (ours) (two-stage: continuity warm-up then enable momentum). Table 2 shows that the velocity magnitude of single-scale variants are consistently inferior, and the two-stage schedule is more stable and accurate than training continuity and momentum jointly from scratch. In addition to architectural and scheduling ablations, we also evaluate the sensitivity of the multi-component loss weights and control-volume radii in Appendix C.4.

## 4.5. Comparison with weak-form PINNs.

Table 1 includes hp-VPINN as a representative variational weak-form PINN baseline. Although weak-form PINNs reduce derivative order and can improve stability on many benchmark PDEs, their standard variational formulations typically rely on predefined subdomains, test-function bases, or domain decomposition. In highly tortuous 3D internal-flow geometries, such weak-form supervision can still be insufficient: the integration regions may not align with the dominant transport pathways, and purely averaged residuals provide limited local guidance for resolving sharp velocity gradients and pressure variations.

MUSA-PINN differs from classical weak-form PINNs in two aspects. First, it is a hybrid strong–weak formulation: the strong-form residual provides pointwise Navier–Stokes supervision, while the weak-form terms add integral conservation constraints rather than replacing the strong-form loss. Indeed, a pure weak-form variant that removes $\mathcal{L}_{sf}$ increases the Diamond-case relative $\ell_2$ errors from 12.46% / 26.55% to 43.92% / 93.81% on velocity magnitude / pres-

sure, confirming that strong-form supervision is not redundant. Second, our weak constraints are imposed through divergence-theorem flux balances on multi-scale clipped control volumes, including skeleton-aware medium-scale volumes that follow channel connectivity. This design provides both local PDE guidance and dense interior conservation supervision, which explains why MUSA-PINN remains stable in complex TPMS channels where hp-VPINN and other baselines degrade.

## 4.6. Performance on More Challenging Regimes

We further stress-test MUSA-PINN under harder physics and geometries to evaluate robustness at higher Reynolds numbers, applicability to industrial components, and empirical capacity scaling.

**Higher-Reynolds-number Gyroid.** We further evaluate the Gyroid channel at higher Reynolds numbers, where stronger inertial effects make the optimization problem more challenging. At $Re = 200$, MUSA-PINN achieves relative $\ell_2$ errors of 14.27% for $|\mathbf{u}|$ and 4.26% for $p$. To directly compare robustness at a more challenging setting, we additionally run all baselines on Gyroid at $Re = 400$ under the same matched training protocol as in Table 1. As shown in Table 3, all baselines degrade substantially at this Reynolds number, with the best baseline, MDPINN-GD, reaching 69.33% velocity error and 61.51% pressure error. In contrast, MUSA-PINN reduces the errors to 21.08% and 8.09%, corresponding to relative improvements of 69.59% in velocity and 86.85% in pressure over the best baseline. These results suggest that MUSA-PINN remains more robust in the higher-Reynolds-number test.

**Industrial geometries: liquid cooling plate and DualMS heat exchanger.** We further test MUSA-PINN on a liquid cooling plate and a DualMS channel to assess robustness on industrial-scale complex geometries. Figures 1, 7 visualizes the flow streamlines of industrial geometries, and the relative $\ell_2$ errors against CFD are reported in the caption. Note

*Table 3.* Comparison on the Gyroid channel at $Re = 400$ under the same matched training protocol. We report relative $\ell_2$ errors for velocity magnitude and pressure. MUSA-PINN retains a clear advantage over all baselines in this higher-Reynolds-number setting.

| Method | Rel. $\ell_2$ on $|\mathbf{u}| \downarrow$ | Rel. $\ell_2$ on $p \downarrow$ |
|---|---|---|
| PINN-PE | 88.02% | 89.37% |
| RoPINN | 76.79% | 94.16% |
| CoPINN | 85.26% | 98.13% |
| MDPINN-GD | 69.33% | 61.51% |
| MUSA-PINN | **21.08%** | **8.09%** |

*Table 4.* **Empirical capacity scaling on the liquid cooling plate.** Relative $\ell_2$ errors of MUSA-PINN with different MLP capacities under the same training protocol. Width $\times$ depth denotes (hidden units per layer) $\times$ (number of hidden layers).

| Width$\times$Depth | #Params | $|\mathbf{u}|$ Rel. $\ell_2 \downarrow$ | $p$ Rel. $\ell_2 \downarrow$ |
|---|---|---|---|
| $256 \times 5$ | 280K | 14.91% | 23.49% |
| $256 \times 8$ | 477K | 14.18% | 11.49% |
| $512 \times 5$ | 1084K | 13.82% | 20.83% |

*Figure 7.* **Flow streamlines on DualMS heat exchanger.** (a) CFD reference, (b) PINN-PE baseline (degrades in this complex geometry), and (c) MUSA-PINN. MUSA-PINN achieves a 20.08% relative $\ell_2$ error in velocity w.r.t. CFD.

the discontinuity in the standard PINN streamlines vs. the smooth flow in MUSA-PINN.

**Empirical capacity scaling.** We keep the training protocol fixed and vary the MLP width and depth to study capacity effects on the liquid cooling plate. Table 4 shows a consistent decrease in relative $\ell_2$ error as capacity increases, indicating that larger networks yield more accurate solutions in this industrial-scale geometry.

## 5. Discussion and Conclusion

In this paper, we presented MUSA-PINN, a novel multi-scale weak-form framework designed to tackle the optimization pathologies of Navier-Stokes equations in topologically complex geometries. By enforcing integral conservation on hierarchically placed control volumes, MUSA-PINN strengthens long-range physical coupling while preserving local fidelity, alleviating the local–global mismatch that often hinders pointwise PINN training. Across TPMS heat-exchanger channels, MUSA-PINN consistently improves accuracy and physical consistency over competitive baselines. We further validate its effectiveness on more realistic internal-flow geometries, including cold-plate and DualMS channels, demonstrating its potential as a scalable volumetric mesh free surrogate for component-level flow simulation.

**Limitations.** MUSA-PINN incurs additional training cost

due to integration on spherical control-volume surfaces, training time details in Appendix D.3. One promising direction is to make the weak-form constraints adaptive and sparse, e.g., activating or reweighting only the most informative control volumes according to residuals, flux imbalance, or uncertainty, while caching surface samples and geometry weights for fixed domains. Moreover, the current control-volume placement is heuristic; learning task-adaptive constraint placement remains an open problem. Future work could learn or adapt the placement from both geometric features, such as skeletons and distance fields, and training signals, such as local conservation errors. We also do not provide a formal convergence theorem; our current theoretical justification is based on the consistency of the strong–weak residuals and Monte Carlo surface quadrature, while a full error analysis remains future work.

**Future Work.** Beyond addressing the above limitations, we identify two extensions for MUSA-PINN. First, we will adapt the framework to transient regimes by constructing spatiotemporal control volumes to enforce conservation over time. Second, we will incorporate energy equations to model conjugate heat transfer, enabling a unified multiphysics framework for end-to-end heat-exchanger optimization.

## Acknowledgements

We thank all reviewers for their valuable comments and constructive suggestions. The authors acknowledge the support of the National Natural Science Foundation of China (NSFC) through the Excellence Research Group Program (Grant No. 52488101). This work was also supported in part by the National Natural Science Foundation of China (Grant Nos. U25A20438 and 62472258).

## Impact Statement

This paper presents work whose goal is to advance the field of machine learning. There are many potential societal consequences of our work, none of which we feel must be specifically highlighted here.

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

## A. Detailed Methodology

### A.1. Reynolds number definition.

The nondimensionalization uses a characteristic length scale $L_0$ and a characteristic velocity scale $U_0$, with

$$Re = \frac{U_0 L_0}{\nu}, \tag{19}$$

where $\nu$ is the kinematic viscosity. In our setting, we take $U_0$ as the prescribed inlet mean velocity, and set $L_0$ to the hydraulic diameter,

$$L_0 := D_h = \frac{4A}{P}, \tag{20}$$

where $A$ is the cross-sectional flow area and $P$ is the corresponding wetted perimeter.

### A.2. Constructing and sampling the clipped boundary $\partial V(\mathbf{c}, r)$

For each local integration subdomain $V(\mathbf{c}, r) = B(\mathbf{c}, r) \cap \Omega$, we need to sample points on the clipped boundary $\partial V(\mathbf{c}, r) = \partial(B(\mathbf{c}, r) \cap \Omega)$. The $\partial V(\mathbf{c}, r)$ decomposes into the spherical part and the clipped geometry part:

$$\partial V(\mathbf{c}, r) = \underbrace{\left(\partial B(\mathbf{c}, r) \cap \Omega\right)}_{\partial V_{\mathrm{sph}}(\mathbf{c}, r)} \cup \underbrace{\left(B(\mathbf{c}, r) \cap \partial \Omega\right)}_{\partial V_{\mathrm{bdry}}(\mathbf{c}, r)}. \tag{21}$$

Accordingly, we construct samples on $\partial V_{\mathrm{sph}}$ and $\partial V_{\mathrm{bdry}}$ via rejection sampling.

**Sphere part** $\partial V_{\mathrm{sph}}(\mathbf{c}, r) = \partial B(\mathbf{c}, r) \cap \Omega$. We first draw $N_B$ sampling points $\mathcal{P}_B = \{\mathbf{y}_i\}_{i=1}^{N_B}$ uniformly on the full sphere surface $\partial B(\mathbf{c}, r)$. We then keep the subset that lies inside the domain:

$$\mathcal{P}_{\mathrm{sph}}(\mathbf{c}, r) = \{\mathbf{y} \in \mathcal{P}_B : \mathbf{y} \in \Omega\}. \tag{22}$$

In our implementation, the inside-domain test is computed by the Embree-accelerated containment query in `trimesh` (`mesh.contains`), which performs a standard inside/outside classification for watertight triangle meshes. For any accepted sphere sample $\mathbf{y} \in \mathcal{P}_{\mathrm{sph}}(\mathbf{c}, r)$, the outward unit normal is analytic:

$$\mathbf{n}(\mathbf{y}) = \frac{\mathbf{y} - \mathbf{c}}{\|\mathbf{y} - \mathbf{c}\|_2}. \tag{23}$$

**Geometry part** $\partial V_{\mathrm{bdry}}(\mathbf{c}, r) = B(\mathbf{c}, r) \cap \partial \Omega$. We represent $\partial \Omega$ as a watertight triangle mesh and draw $N_\Omega$ sampling points $\mathcal{P}_\Omega = \{\mathbf{x}_i\}_{i=1}^{N_\Omega}$ approximately uniformly with respect to surface area on $\partial \Omega$. We then keep the subset that lies inside the ball:

$$\mathcal{P}_{\mathrm{bdry}}(\mathbf{c}, r) = \{\mathbf{x} \in \mathcal{P}_\Omega : \|\mathbf{x} - \mathbf{c}\|_2 \le r\}. \tag{24}$$

For any accepted geometry sample $\mathbf{x} \in \mathcal{P}_{\mathrm{bdry}}(\mathbf{c}, r)$, the outward normal $\mathbf{n}(\mathbf{x})$ is provided by the mesh.

**Reuse for Monte Carlo integration.** The accepted sample sets $\mathcal{P}_{\mathrm{sph}}(\mathbf{c}, r)$ and $\mathcal{P}_{\mathrm{bdry}}(\mathbf{c}, r)$ are used both for (i) estimating the component areas $A_{\mathrm{sph}}$ and $A_{\mathrm{bdry}}$ (Appendix A.4) and (ii) evaluating the corresponding surface integrals by Monte Carlo quadrature (Appendix A.3), thereby avoiding additional sampling overhead.

### A.3. Monte Carlo estimation of surface integrals.

The weak-form residuals $R_c(\mathbf{c}, r)$ and $\mathbf{R}_m(\mathbf{c}, r)$ involve surface integrals over the boundary $\partial V(\mathbf{c}, r)$. We approximate these integrals via Monte Carlo surface sampling. For any integrand $g(\mathbf{x})$ defined on $\partial V(\mathbf{c}, r)$, we use

$$\oint_{\partial V(\mathbf{c}, r)} g(\mathbf{x}) \, dS \approx A(\mathbf{c}, r) \cdot \frac{1}{K} \sum_{k=1}^{K} g(\mathbf{x}_k), \tag{25}$$

where $\{\mathbf{x}_k\}_{k=1}^{K}$ are samples approximately uniform on $\partial V(\mathbf{c}, r)$, and $A(\mathbf{c}, r) = \int_{\partial V(\mathbf{c}, r)} dS$ is the boundary area. For vector-valued integrands, we apply the estimator component-wise.

In practice, $\partial V(\mathbf{c}, r)$ can be decomposed into two components:

$$\partial V(\mathbf{c}, r) = \partial V_{\text{sph}} \cup \partial V_{\text{bdry}}, \tag{26}$$

with $\partial V_{\text{sph}} = \partial B(\mathbf{c}, r) \cap \Omega$ and $\partial V_{\text{bdry}} = B(\mathbf{c}, r) \cap \partial \Omega$. We sample points on each component and combine them with area weights:

$$\oint_{\partial V} g \, dS \approx A_{\text{sph}} \cdot \frac{1}{K_{\text{sph}}} \sum_{k=1}^{K_{\text{sph}}} g(\mathbf{x}_k^{\text{sph}}) + A_{\text{bdry}} \cdot \frac{1}{K_{\text{bdry}}} \sum_{k=1}^{K_{\text{bdry}}} g(\mathbf{x}_k^{\text{bdry}}), \tag{27}$$

where $K_{\text{sph}} + K_{\text{bdry}} = K$ and $A_{\text{sph}} + A_{\text{bdry}} = A(\mathbf{c}, r)$. On $\partial V_{\text{sph}}$, the outward unit normal is analytic: $\mathbf{n} = (\mathbf{x} - \mathbf{c})/\|\mathbf{x} - \mathbf{c}\|$. On $\partial V_{\text{bdry}}$, $\mathbf{n}$ is provided by the geometry. The areas $A_{\text{sph}}$ and $A_{\text{bdry}}$ are estimated by rejection sampling on the sphere and the geometry surface, respectively (Appendix A.4).

Applying the Monte Carlo estimator to our weak-form residuals amounts to sampling the boundary flux integrands on $\partial V(\mathbf{c}, r)$.

For the continuity residual in Eq. (8), the scalar integrand is

$$g_c(\mathbf{x}) = \mathbf{u}_\theta(\mathbf{x}) \cdot \mathbf{n}(\mathbf{x}) \in \mathbb{R}. \tag{28}$$

For the momentum residual in Eq. (12), the integrand is the vector

$$\mathbf{g}_m(\mathbf{x}) = \mathbf{F}_\theta(\mathbf{x})\mathbf{n}(\mathbf{x}) \in \mathbb{R}^3, \tag{29}$$

where $\mathbf{F}_\theta = \mathbf{u}_\theta \otimes \mathbf{u}_\theta + p_\theta \mathbf{I} - \frac{1}{Re}\nabla \mathbf{u}_\theta$ is the network-induced flux tensor, and $\mathbf{u}_\theta$, $p_\theta$ are predicted by the network $f_\theta$.

### A.4. Estimating $A_{\text{sph}}$ and $A_{\text{bdry}}$ by Rejection Sampling

Recall the boundary decomposition $\partial V(\mathbf{c}, r) = \partial V_{\text{sph}} \cup \partial V_{\text{bdry}}$, where $\partial V_{\text{sph}} = \partial B(\mathbf{c}, r) \cap \Omega$ and $\partial V_{\text{bdry}} = B(\mathbf{c}, r) \cap \partial \Omega$. To combine Monte Carlo estimates from the two components, we require the component areas $A_{\text{sph}}$ and $A_{\text{bdry}}$. We estimate them via rejection sampling using simple membership tests.

**Spherical component area $A_{\text{sph}}$.** Following Appendix A.2, we draw $N_B$ sampling points $\mathcal{P}_B = \{\mathbf{y}_i\}_{i=1}^{N_B}$ uniformly on $\partial B(\mathbf{c}, r)$ and obtain the accepted subset $\mathcal{P}_{\text{sph}}(\mathbf{c}, r) = \{\mathbf{x}_k^{\text{sph}}\}_{k=1}^{K_{\text{sph}}}$ via the inside-domain test in Eq. (22). The acceptance ratio is estimated as

$$\widehat{p}_{\text{sph}} = \frac{K_{\text{sph}}}{N_B} = \frac{1}{N_B} \sum_{i=1}^{N_B} \mathbb{I}[\mathbf{y}_i \in \Omega]. \tag{30}$$

Since the surface area of $\partial B(\mathbf{c}, r)$ equals $4\pi r^2$, we estimate

$$\widehat{A}_{\text{sph}} = 4\pi r^2 \cdot \widehat{p}_{\text{sph}}. \tag{31}$$

**Boundary component area $A_{\text{bdry}}$.** Following Appendix A.2, we draw $N_\Omega$ sampling points $\mathcal{P}_\Omega = \{\mathbf{z}_i\}_{i=1}^{N_\Omega}$ approximately uniformly (w.r.t. surface area) on $\partial \Omega$ and obtain the accepted subset $\mathcal{P}_{\text{bdry}}(\mathbf{c}, r) = \{\mathbf{x}_k^{\text{bdry}}\}_{k=1}^{K_{\text{bdry}}}$ via the inside-ball test in Eq. (24). The acceptance ratio is estimated as

$$\widehat{p}_{\text{bdry}} = \frac{K_{\text{bdry}}}{N_\Omega} = \frac{1}{N_\Omega} \sum_{i=1}^{N_\Omega} \mathbb{I}[\|\mathbf{z}_i - \mathbf{c}\|_2 \leq r]. \tag{32}$$

Let $A(\partial \Omega)$ denote the total surface area of $\partial \Omega$. We estimate the clipped boundary area as

$$\widehat{A}_{\text{bdry}} = A(\partial \Omega) \cdot \widehat{p}_{\text{bdry}}. \tag{33}$$

**Remarks.** (i) The estimators above are consistent under area-uniform sampling on the corresponding surfaces. (ii) In our implementation, the same samples used to estimate $\widehat{A}_{\text{sph}}$ and $\widehat{A}_{\text{bdry}}$ can also be reused for evaluating surface integrals on each component, thereby avoiding additional sampling overhead.

## A.5. Radius selection.

Our goal is to choose the three radii $(r_L, r_M, r_S)$ in a geometry-aware and reproducible way, so that (i) large subdomains provide long-range coupling, (ii) medium subdomains match the channel scale along the skeleton, and (iii) small subdomains refine local geometric details while keeping the Monte Carlo estimator stable.

**Geometry scales.** Let the axis-aligned bounding box of $\Omega$ have side lengths $(\Delta x, \Delta y, \Delta z)$. Denote the sorted side lengths by $d_1 \leq d_2 \leq d_3$. To characterize the channel thickness, we use the distance-to-boundary (local inscribed radius)

$$\rho(\mathbf{x}) = \text{dist}(\mathbf{x}, \partial\Omega), \qquad \mathbf{x} \in \Omega, \tag{34}$$

which is evaluated either from a signed distance field (SDF) or nearest-boundary queries. We compute $\rho(\mathbf{c})$ on skeleton centers $\mathbf{c} \in \mathcal{C}_M$ to obtain a robust estimate of the "wide-channel" scale:

$$\hat{\rho}_M = Q_{0.9}\big(\{\rho(\mathbf{c})\}_{\mathbf{c} \in \mathcal{C}_M}\big), \tag{35}$$

where $Q_{0.9}(\cdot)$ denotes the 90% quantile. Using a high quantile behaves similarly to the maximum channel radius, but is more robust to outliers.

**Radius rules.** We set the radii by the following rules:

$$r_L = \alpha_L \cdot \frac{1}{2}\sqrt{d_1^2 + d_2^2}, \tag{36}$$

$$r_M = \alpha_M \cdot \hat{\rho}_M, \tag{37}$$

$$r_S = \beta \cdot r_M. \tag{38}$$

Eq. (36) chooses $r_L$ as an inflated circumradius of the rectangle formed by the two smallest box extents, which avoids overly large subdomains in highly anisotropic geometries while still enabling domain-level coverage and long-range flux coupling. Eq. (37) ties $r_M$ to the channel thickness along the skeleton so that skeleton-centered subdomains cover the transport pathways. Eq. (38) sets a nested refinement scale relative to the medium one.

In our experiments, we use fixed coefficients across all experiments:

$$\alpha_L = \alpha_M = 1.4, \qquad \beta = 0.5. \tag{39}$$

Here $\alpha_L, \alpha_M > 1$ are mild inflation factors that increase overlap and compensate for geometric truncation, and $\beta$ controls the refinement ratio between medium and small scales. We found performance to be insensitive to moderate changes of these coefficients due to the complementary multi-scale coverage.

## A.6. Boundary Conditions.

We enforce Dirichlet-type inlet velocity, outlet pressure, and no-slip wall conditions using mean-squared penalty terms on boundary point sets. Let $\mathcal{X}_{\text{bc}} \subset \partial\Omega$ denote the sampled boundary points. We partition it according to boundary types:

$$\mathcal{X}_{\text{in}} = \mathcal{X}_{\text{bc}} \cap \Gamma_{\text{in}}, \quad \mathcal{X}_{\text{out}} = \mathcal{X}_{\text{bc}} \cap \Gamma_{\text{out}}, \quad \mathcal{X}_{\text{w}} = \mathcal{X}_{\text{bc}} \cap \Gamma_{\text{w}}. \tag{40}$$

For each $\mathbf{x} \in \mathcal{X}_{\text{bc}}$, the network prediction is $(\mathbf{u}_\theta(\mathbf{x}), p_\theta(\mathbf{x})) = f_\theta(\mathbf{x})$. The prescribed boundary values are given by $\mathbf{u}_{\text{in}}(\mathbf{x})$ on $\Gamma_{\text{in}}$ and $p_{\text{out}}(\mathbf{x})$ on $\Gamma_{\text{out}}$.

The inlet-velocity penalty is

$$\mathcal{L}_{\text{in}} = \frac{1}{|\mathcal{X}_{\text{in}}|} \sum_{\mathbf{x} \in \mathcal{X}_{\text{in}}} \|\mathbf{u}_\theta(\mathbf{x}) - \mathbf{u}_{\text{in}}(\mathbf{x})\|_2^2. \tag{41}$$

The outlet-pressure penalty is

$$\mathcal{L}_{\text{out}} = \frac{1}{|\mathcal{X}_{\text{out}}|} \sum_{\mathbf{x} \in \mathcal{X}_{\text{out}}} |p_\theta(\mathbf{x}) - p_{\text{out}}(\mathbf{x})|^2. \tag{42}$$

The no-slip wall penalty is

$$\mathcal{L}_{\text{w}} = \frac{1}{|\mathcal{X}_{\text{w}}|} \sum_{\mathbf{x} \in \mathcal{X}_{\text{w}}} \|\mathbf{u}_\theta(\mathbf{x})\|_2^2. \tag{43}$$

Finally, the boundary-condition loss used in the main objective is

$$\mathcal{L}_{\text{bc}} = \lambda_{\text{in}}\mathcal{L}_{\text{in}} + \lambda_{\text{out}}\mathcal{L}_{\text{out}} + \lambda_{\text{w}}\mathcal{L}_{\text{w}}. \tag{44}$$

## B. Detailed Experimental Setup and Ground Truth Generation

In this section, we provide the precise definitions of the geometric configurations, physical parameters, and numerical protocols used to generate the high-fidelity ground truth data via Ansys CFX.

### B.1. Geometric Definitions of TPMS Structures

The porous media geometries used in this study are defined by Triply Periodic Minimal Surfaces (TPMS). The implicit level-set equations approximation for the Primitive ($P$), Gyroid ($G$), Diamond ($D$), and IWP structures are given as follows:

$$\Phi_P(x, y, z) = \cos(2\pi x) + \cos(2\pi y) + \cos(2\pi z) = 0, \tag{45}$$

$$\Phi_G(x, y, z) = \sin(2\pi x)\cos(2\pi y) + \sin(2\pi y)\cos(2\pi z) + \sin(2\pi z)\cos(2\pi x) = 0, \tag{46}$$

$$\Phi_D(x, y, z) = \sin(2\pi x)\sin(2\pi y)\sin(2\pi z) + \sin(2\pi x)\cos(2\pi y)\cos(2\pi z)$$
$$+ \cos(2\pi x)\sin(2\pi y)\cos(2\pi z) + \cos(2\pi x)\cos(2\pi y)\sin(2\pi z) = 0, \tag{47}$$

$$\Phi_{IWP}(x, y, z) = 2\left(\cos(2\pi x)\cos(2\pi y) + \cos(2\pi y)\cos(2\pi z) + \cos(2\pi z)\cos(2\pi x)\right)$$
$$- \left(\cos(4\pi x) + \cos(4\pi y) + \cos(4\pi z)\right) = 0. \tag{48}$$

Fig. 8 visualizes the four TPMS geometries and our domain construction: we tile five unit cells along the streamwise $x$-direction, resulting in $\Omega = [0, 5L] \times [0, L] \times [0, L]$ with unit-cell length $L = 1$.

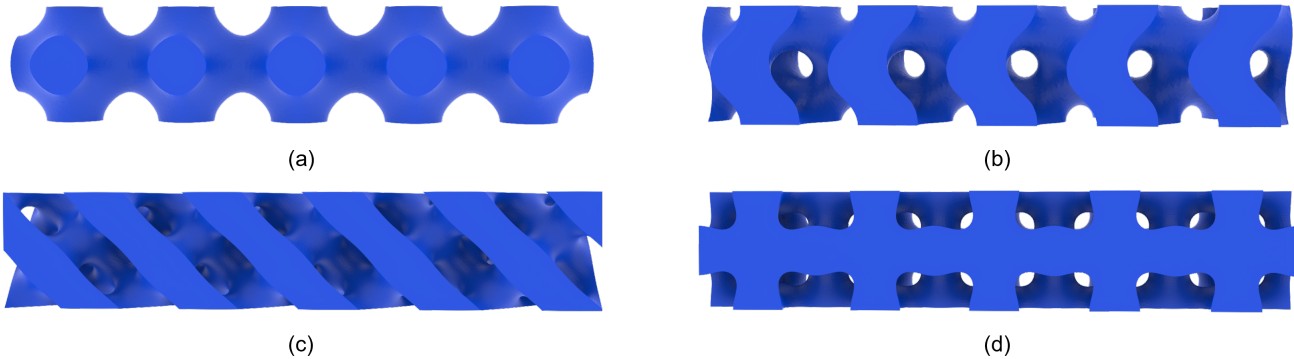

(a)  (b)

(c)  (d)

*Figure 8.* **TPMS heat-exchanger channels used in our experiments.** The domain is constructed by tiling five unit cells along the streamwise $x$-direction, yielding $\Omega = [0, 5L] \times [0, L] \times [0, L]$ with unit-cell length $L = 1$. Panels (a)–(d) show *Primitive (P)*, *Gyroid (G)*, *Diamond (D)*, and *IWP*, respectively.

### B.2. Simulation Settings and Numerical Solver Configuration

**Governing Equations and Boundary Conditions**   We simulate the steady-state flow of an incompressible Newtonian fluid governed by the Navier-Stokes equations:

$$\nabla \cdot \mathbf{u} = 0, \tag{49}$$

$$(\mathbf{u} \cdot \nabla)\mathbf{u} = -\frac{1}{\rho}\nabla p + \nu\nabla^2\mathbf{u}, \tag{50}$$

where $\mathbf{u}$ is the velocity vector, $p$ is pressure, $\rho$ is density, and $\nu$ is kinematic viscosity. The simulation is characterized by a Reynolds number $Re = \frac{U_{in}L}{\nu} = 100$, ensuring the flow remains in the laminar regime.

**Numerical Solver Configuration**   High-fidelity ground truth data were generated using Ansys CFX, a commercial CFD solver based on the finite volume method (FVM). The specific solver settings are detailed as follows:

- **Mesh Generation:** An unstructured tetrahedral mesh was generated to discretize the computational domain. To accurately capture the boundary layer gradients, prismatic inflation layers were applied near the wall surfaces. The

boundary layer mesh consisted of 5 layers with a first layer height of 0.002 and a growth rate of 1.2. A global element size of 0.02 was applied, with a refined maximum size of 0.01 near the boundaries. This configuration resulted in a high-resolution grid containing approximately $10^7$ (tens of millions) elements.

- **Discretization Scheme:** A High-Resolution advection scheme was employed for the spatial discretization of momentum and mass equations to ensure both numerical stability and second-order equivalent accuracy.

- **Convergence Criteria:** The simulations were subjected to strict convergence criteria. The iterative process was terminated only when the root-mean-square (RMS) residuals for all governing equations dropped below $1 \times 10^{-8}$, ensuring a fully converged and high-fidelity solution.

- **Inlet/Outlet Extensions:** Straight pipe extensions were added at both the inlet and outlet. The inlet extension ensured a developed inflow before entering the TPMS region, while the outlet extension minimized outlet boundary effects and prevented backflow/recirculation near the outlet from contaminating the flow in the TPMS region. The extension length was set to three TPMS unit-cell lengths.

## C. Network Architecture and Hyperparameter Values

In this section, we provide the formal definition of the neural network architecture, evaluate the impact of different positional encoding strategies, and detail the hyperparameter configurations to facilitate reproducibility.

### C.1. Network Architecture

The proposed physics-informed framework employs a fully connected feed-forward neural network (MLP) augmented with a Fourier feature mapping layer. This architecture is designed to mitigate the spectral bias inherent in standard coordinate-based MLPs.

**Fourier Feature Embedding.** Let $\mathbf{x} \in \Omega \subset \mathbb{R}^3$ denote the input spatial coordinates. We project $\mathbf{x}$ into a high-dimensional feature space using a deterministic Fourier mapping $\gamma : \mathbb{R}^3 \rightarrow \mathbb{R}^{2m}$. Unlike standard Random Fourier Features (RFF) where frequencies are sampled from a Gaussian distribution, we employ linearly spaced frequency bands to ensure uniform spectral coverage within the target bandwidth. The embedding is defined as:

$$\gamma(\mathbf{x}) = [\sin(2\pi\mathbf{B}\mathbf{x}), \cos(2\pi\mathbf{B}\mathbf{x})]^{\top}, \tag{51}$$

where $\mathbf{B} \in \mathbb{R}^{m \times 3}$ represents the frequency matrix. The frequencies are initialized linearly within the range $[f_{min}, f_{max}]$ to capture multi-scale flow features. In our final model, we set the embedding dimension to $2m = 60$.

**Backbone and Output Layers.** The embedded features $\gamma(\mathbf{x})$ serve as the input to an MLP comprising $D$ hidden layers with $W$ neurons per layer. We utilize the Tanh activation function $\sigma(\cdot)$ to guarantee the higher-order differentiability required for computing PDE derivatives. The forward propagation is given by:

$$\mathbf{h}_0 = \gamma(\mathbf{x}), \tag{52}$$
$$\mathbf{h}_k = \sigma(\mathbf{W}_k\mathbf{h}_{k-1} + \mathbf{b}_k), \quad k = 1, \ldots, D. \tag{53}$$

The final output layer projects the hidden features $\mathbf{h}_D$ to the physical variables of interest, namely the velocity vector $\mathbf{u} = (u, v, w)$ and pressure $p$, via a linear transformation without activation:

$$[\mathbf{u}, p] = \mathbf{W}_{out}\mathbf{h}_D + \mathbf{b}_{out}. \tag{54}$$

Network weights are initialized using the Glorot scheme.

### C.2. Position Encoding Comparison

To identify the optimal spectral mapping strategy for flow fields, we conducted a systematic evaluation of four embedding configurations on the Gyroid geometry ($Re = 100$):

(i) **Standard MLP**: Direct coordinate input without embedding ($\gamma(\mathbf{x}) = \mathbf{x}$), serving as the baseline.

(ii) **Gaussian Random Fourier Feature**: Frequencies sampled from a Gaussian distribution $\mathbf{B} \sim \mathcal{N}(0, \sigma^2)$, introducing stochasticity.

(iii) **Exponential Fourier Feature**: Deterministic frequencies spaced geometrically (i.e., $f_k = \alpha^k$), a standard setting in neural rendering to capture multi-resolution features.

(iv) **Linear Fourier Feature (Ours)**: Deterministic frequencies spaced linearly within the target bandwidth $[f_{min}, f_{max}]$, ensuring uniform spectral coverage.

**Quantitative Results.** Table 5 summarizes the convergence performance. The Standard MLP suffers from spectral bias, failing to resolve sharp boundary layers. Gaussian Random Fourier Feature improves accuracy but exhibits variance due to random sampling. Exponential Fourier Feature effectively captures global structures but empirically struggles with mid-frequency transition regions in our fluid benchmarks. In contrast, Linear Fourier Feature achieves the lowest absolute $\ell_2$ error, suggesting that uniform spectral coverage is critical for resolving the continuous multi-scale features inherent in Navier-Stokes dynamics.

*Table 5.* Comparative analysis of positional encoding strategies. **Linear-FF** demonstrates superior convergence and final accuracy compared to RFF and Exponential FF mappings.

| Encoding Strategy | Type | Rel. $\ell_2$ on $|\mathbf{u}|$ ↓ | Rel. $\ell_2$ on $p$ ↓ |
|---|---|---|---|
| Standard MLP | Identity | 70.47% | 33.93% |
| Gaussian RFF | Stochastic | 18.64% | 4.74% |
| Exponential FF | Deterministic | 6.92% | 30.73% |
| **Linear-FF (Ours)** | **Deterministic** | **3.76%** | **5.36%** |

### C.3. Hyperparameter Values

We summarize all hyperparameters used in our experiments for reproducibility, including boundary-condition sampling, strong-form collocation sampling, local subdomain sampling (large/medium/small balls), Monte Carlo surface sampling on $\partial V(\mathbf{c}, r)$, optimization, and loss weights in Table 6.

All experiments in Table 1 were conducted on a single NVIDIA A40 GPU using PyTorch, and we use a fully-connected MLP to parameterize the solution fields. Given an input 3D coordinate $\mathbf{x} \in \mathbb{R}^3$, we first apply a linear Fourier feature mapping to obtain a 60-dimensional embedding $\phi(\mathbf{x}) \in \mathbb{R}^{60}$ with frequencies bounded in $[1, 2.5]$. The embedding is then fed into an MLP with 5 hidden layers of width 256. The network outputs a 4-dimensional vector $\hat{\mathbf{y}}(\mathbf{x}) = [u(\mathbf{x}), v(\mathbf{x}), w(\mathbf{x}), p(\mathbf{x})]$, corresponding to the three velocity components and pressure.

### C.4. Hyperparameter Sensitivity

To evaluate the robustness of the multi-component loss weights, we conduct a representative group-wise perturbation study on the Diamond TPMS case. Starting from the default configuration in Table 6, we multiply one group of hyperparameters by a scale factor while keeping all other settings unchanged. We consider boundary-condition weights, strong-form continuity and momentum weights, weak-form continuity and momentum weights, and the three control-volume radii.

Table 7 shows that the default configuration achieves the best overall trade-off between velocity and pressure accuracy. Some perturbations improve pressure reconstruction, e.g., increasing certain mass-conservation weights, but they often lead to noticeably worse velocity errors. This indicates that the multi-component objective involves a velocity–pressure trade-off rather than a single uniformly optimal scaling for every metric. Since the main goal of MUSA-PINN is accurate flow and transport reconstruction, we prioritize velocity accuracy while keeping pressure error within a reasonable range. Importantly, all main results are obtained using the same fixed default weights, without per-geometry retuning. Similar parameter magnitudes were also used for larger industrial geometries such as the liquid-cooling plate.

## D. More Experiments Results

### D.1. Longitudinal Flow Fidelity

To qualitatively assess how well each method preserves the streamwise flow evolution from inlet to outlet, we visualize a longitudinal mid-plane slice in the Gyroid geometry. Specifically, for the 5-period domain $\Omega = [0, 5] \times [0, 1] \times [0, 1]$, we

*Table 6.* **Training hyperparameters for MUSA-PINN.** Numbers of boundary-condition samples and interior collocation points for strong-form residuals, multi-scale spherical subdomain configuration $(N_L, N_M, N_S; r_L, r_M, r_S)$, and optimization settings including the two-stage schedule $(T_{\text{switch}})$ and loss weights.

| Item | Value |
|---|---|
| Inlet samples $|\mathcal{X}_{\text{in}}|$ | 1000 |
| Outlet samples $|\mathcal{X}_{\text{out}}|$ | 1000 |
| Wall (no-slip) samples $|\mathcal{X}_{\text{w}}|$ | 10000 |
| Interior PDE collocation points | 250000 |
| Numbers of Large subdomains $N_L$ | 40 |
| Numbers of Medium subdomains $N_M$ | 200 |
| Numbers of Small subdomains $N_S$ | 500 |
| Large-ball radius $r_L$ | 1.0 |
| Medium-ball radius $r_M$ | 0.5 |
| Small-ball radius $r_S$ | 0.25 |
| Optimizer | SOAP |
| Learning rate (stage 1) | 1e-3 |
| Learning rate (stage 2) | 1e-6 |
| Total epoch | 9000 |
| Stage transition $T_{switch}$ | 7000 |
| Inlet BC weight $\lambda_{\text{in}}$ | 10 |
| Outlet BC weight $\lambda_{\text{out}}$ | 10 |
| No-slip wall BC weight $\lambda_{\text{w}}$ | 10 |
| Strong-form continuity weight $\lambda_c$ | 10 |
| Strong-form momentum weight $\lambda_m$ | 0.1 |
| Weak-form continuity (large) weight $\lambda_c^L$ | 100 |
| Weak-form continuity (medium) weight $\lambda_c^M$ | 100 |
| Weak-form continuity (small) weight $\lambda_c^S$ | 100 |
| Weak-form momentum (large) weight $\lambda_m^L$ | 4 |
| Weak-form momentum (medium) weight $\lambda_m^M$ | 25 |
| Weak-form momentum (small) weight $\lambda_m^S$ | 100 |

render the streamwise velocity component $u_x(\mathbf{x})$ on the plane

$$\Pi_z = \{(x, y, z) \in \Omega : \ z = 0.5\}. \tag{55}$$

This plane spans the full streamwise extent ($x \in [0, 5]$) and provides a compact view of how the flow develops along tortuous passages. Figure 9 compares the resulting longitudinal slices across different methods.

### D.2. Local Flow Fidelity

In the main paper, due to space constraints, we only visualize the *scalar* velocity field for competing methods, while omitting the component-wise behaviors. However, matching $|\mathbf{u}|$ alone can hide important directional errors, such as spurious cross-flow, incorrect recirculation orientation, or component-wise bias. To provide a more fine-grained qualitative assessment, we further visualize the three velocity components $\mathbf{u} = (u, v, w)$ for all methods on the same set of slices. Specifically, Figures. 10, 11, 12, 13, 14, and 15 correspond to PINN-PE, RoPINN, CoPINN, MDPINN-GD, hp-VPINN and MUSA-PINN, respectively, comparing the velocity at $x = 0.5$ with the ground truth.

### D.3. Training time

To provide a fair and reproducible comparison of computational overhead, we report both wall-clock training time for all methods in Table 8.

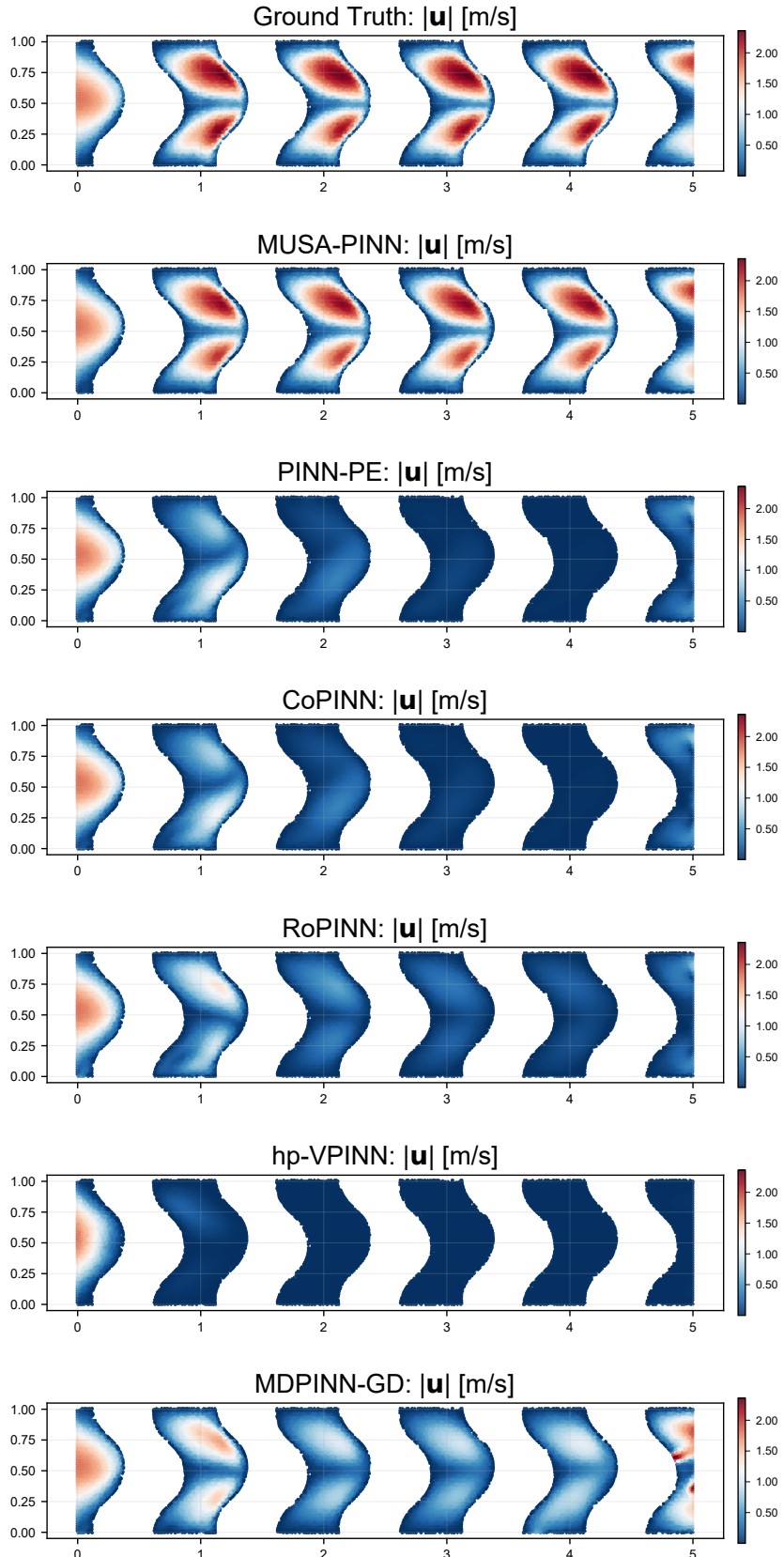

*Figure 9.* **Local flow fidelity in the Gyroid geometry.**

*Table 7.* Hyperparameter sensitivity on the Diamond TPMS case. We perturb one group of hyperparameters at a time by a multiplicative scale factor while keeping all other settings fixed. The default setting provides the best overall velocity–pressure trade-off. Some perturbations improve pressure accuracy but degrade velocity accuracy, indicating a trade-off between pressure reconstruction and flow-field fidelity.

| Weight group | Scale factor | Rel. $\ell_2$ on $|\mathbf{u}|$ / $p$ |
|---|---|---|
| Default | $1\times$ | 12.46% / 26.55% |
| Boundary conditions | $0.1\times$ | 21.38% / 2.70% |
| | $0.2\times$ | 18.31% / 20.78% |
| | $5\times$ | 22.07% / 30.45% |
| | $10\times$ | 30.21% / 52.97% |
| Strong-form mass | $0.1\times$ | 28.16% / 4.01% |
| | $0.2\times$ | 16.69% / 21.18% |
| | $5\times$ | 20.73% / 10.32% |
| | $10\times$ | 24.10% / 3.01% |
| Strong-form momentum | $0.1\times$ | 23.00% / 69.51% |
| | $0.2\times$ | 20.91% / 15.89% |
| | $5\times$ | 15.05% / 27.69% |
| | $10\times$ | 18.77% / 17.95% |
| Weak-form mass | $0.1\times$ | 16.17% / 24.41% |
| | $0.2\times$ | 15.74% / 26.35% |
| | $5\times$ | 20.35% / 7.64% |
| | $10\times$ | 22.54% / 3.69% |
| Weak-form momentum | $0.1\times$ | 16.66% / 24.46% |
| | $0.2\times$ | 15.74% / 26.35% |
| | $5\times$ | 20.35% / 7.65% |
| | $10\times$ | 16.31% / 25.21% |
| Sphere radius | $0.67\times$ | 21.20% / 5.73% |
| | $0.77\times$ | 17.03% / 23.75% |
| | $1.3\times$ | 17.80% / 22.84% |
| | $1.5\times$ | 24.21% / 9.99% |

*Table 8.* **Efficiency comparison.** Convergence iterations and training time.

| Method | Conv. iters | Time (h) |
|---|---|---|
| PINN-PE | 15000 | 10.22 |
| RoPINN | 7500 | 12.35 |
| CoPINN | 15000 | 10.256 |
| MDPINN-GD | 45000 | 38.37 |
| MUSA-PINN (Stage 1) | 7000 | 7.05 |
| MUSA-PINN (Stage 2) | 2000 | 18.79 |

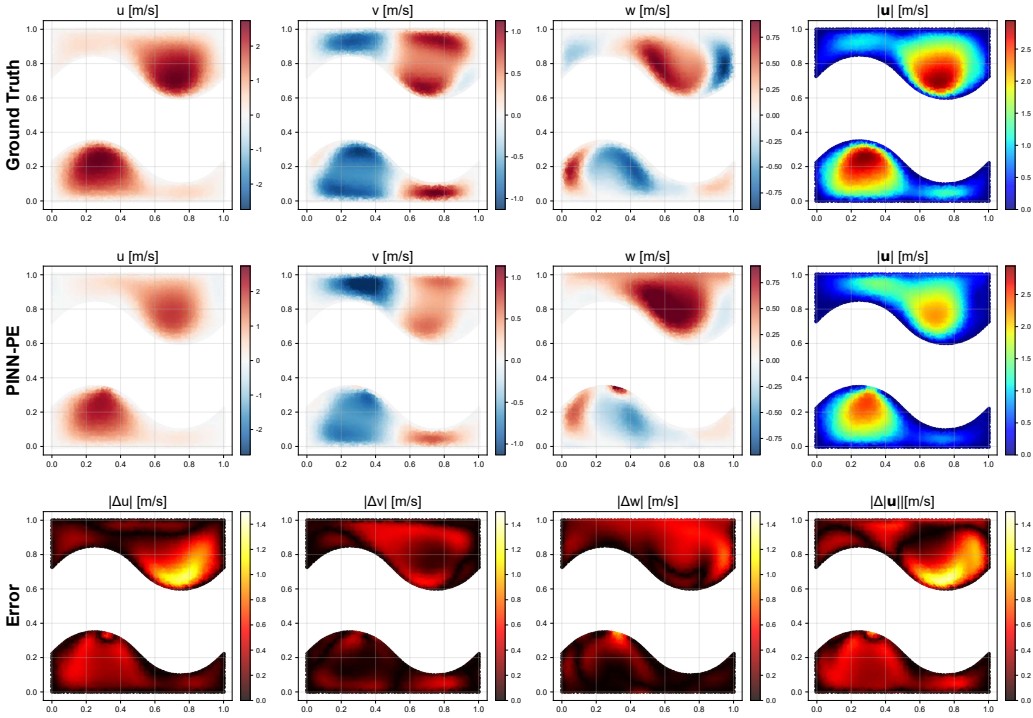

*Figure 10.* **Local flow fidelity at** $x = 0.5$ **(PINN-PE).** Component-wise velocity slices $(u, v, w)$ predicted by PINN-PE compared with the ground truth (GT) on the plane $x = 0.5$.

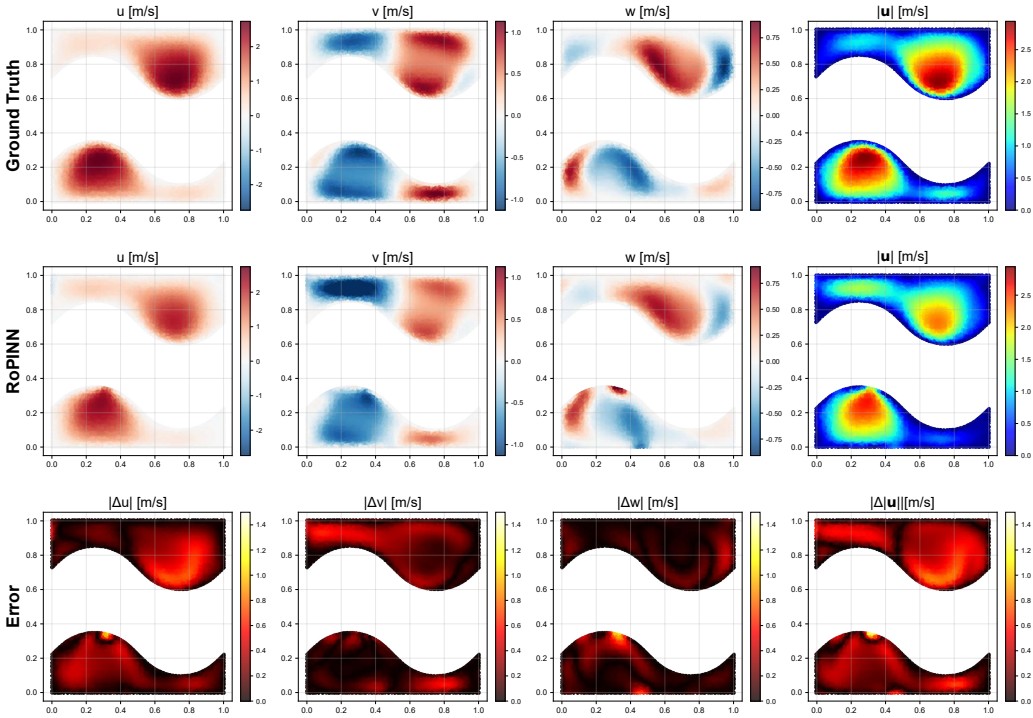

*Figure 11.* **Local flow fidelity at** $x = 0.5$ **(RoPINN).** Component-wise velocity slices $(u, v, w)$ predicted by RoPINN compared with the ground truth (GT) on the plane $x = 0.5$.

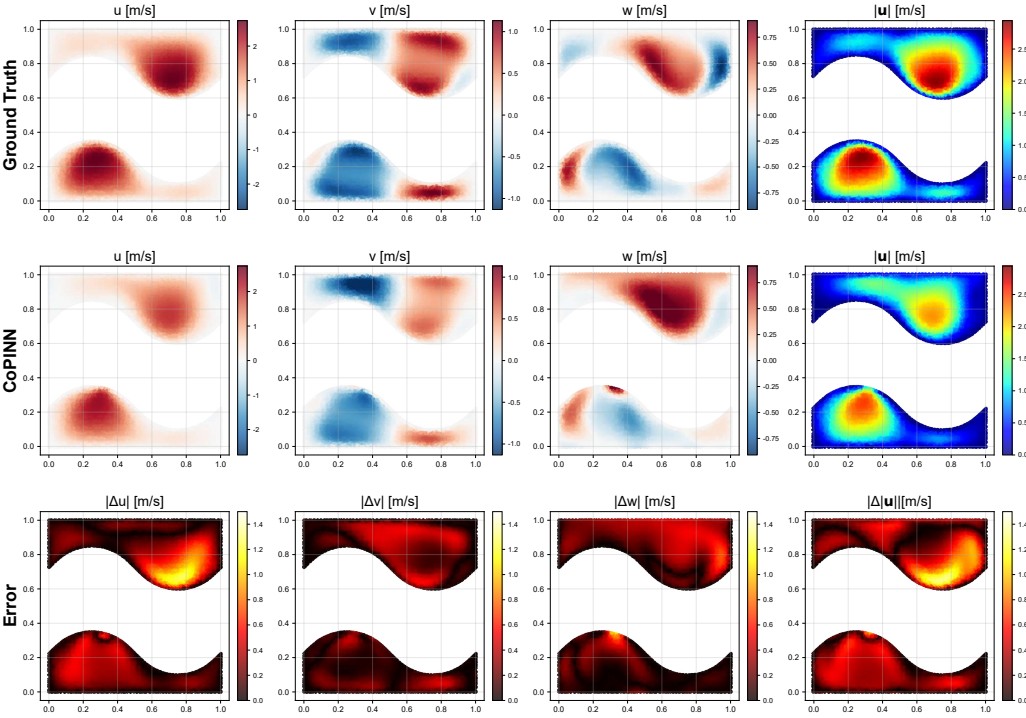

*Figure 12.* **Local flow fidelity at** $x = 0.5$ **(CoPINN).** Component-wise velocity slices $(u, v, w)$ predicted by CoPINN compared with the ground truth (GT) on the plane $x = 0.5$.

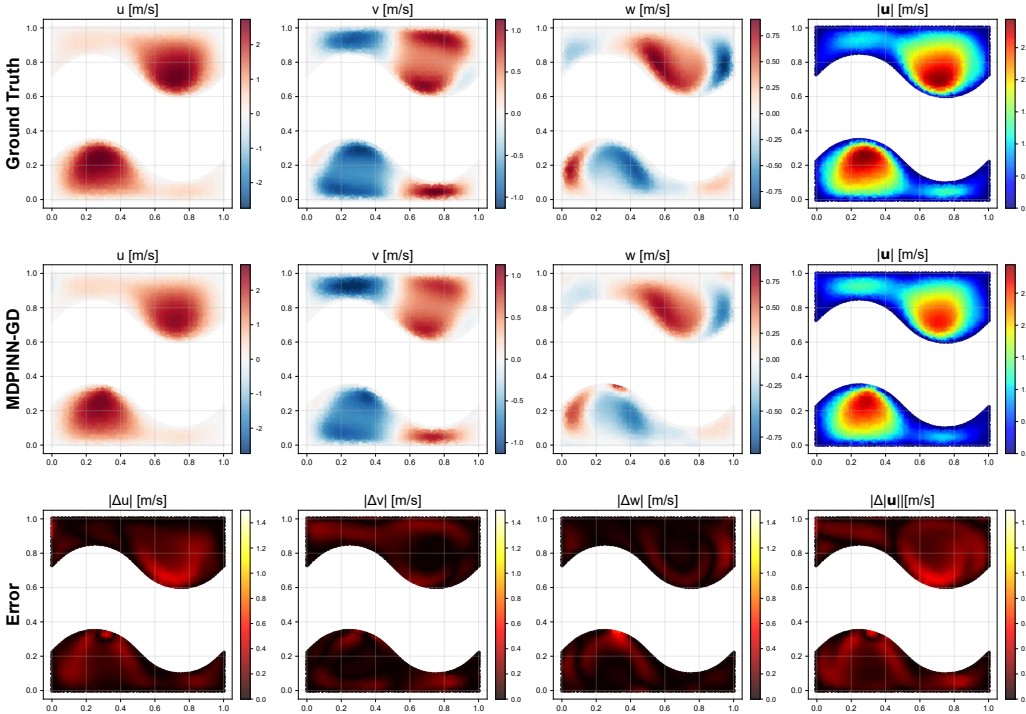

*Figure 13.* **Local flow fidelity at** $x = 0.5$ **(MDPINN-GD).** Component-wise velocity slices $(u, v, w)$ predicted by MDPINN-GD compared with the ground truth (GT) on the plane $x = 0.5$.

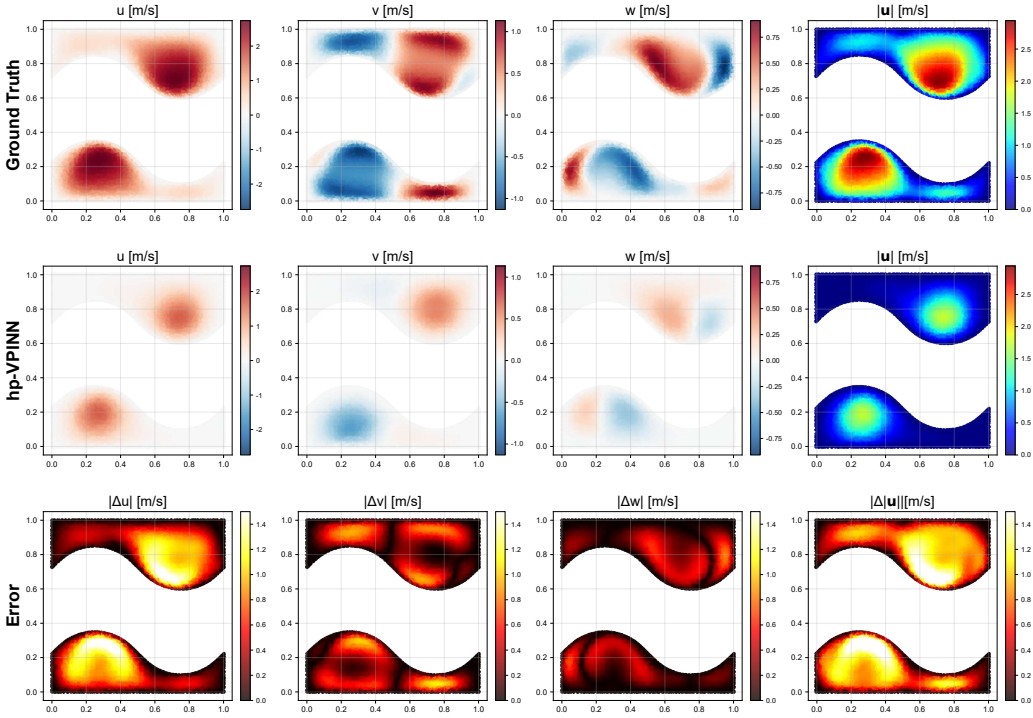

*Figure 14.* **Local flow fidelity at** $x = 0.5$ **(hp-VPINN).** Component-wise velocity slices $(u, v, w)$ predicted by hp-VPINN compared with the ground truth (GT) on the plane $x = 0.5$.

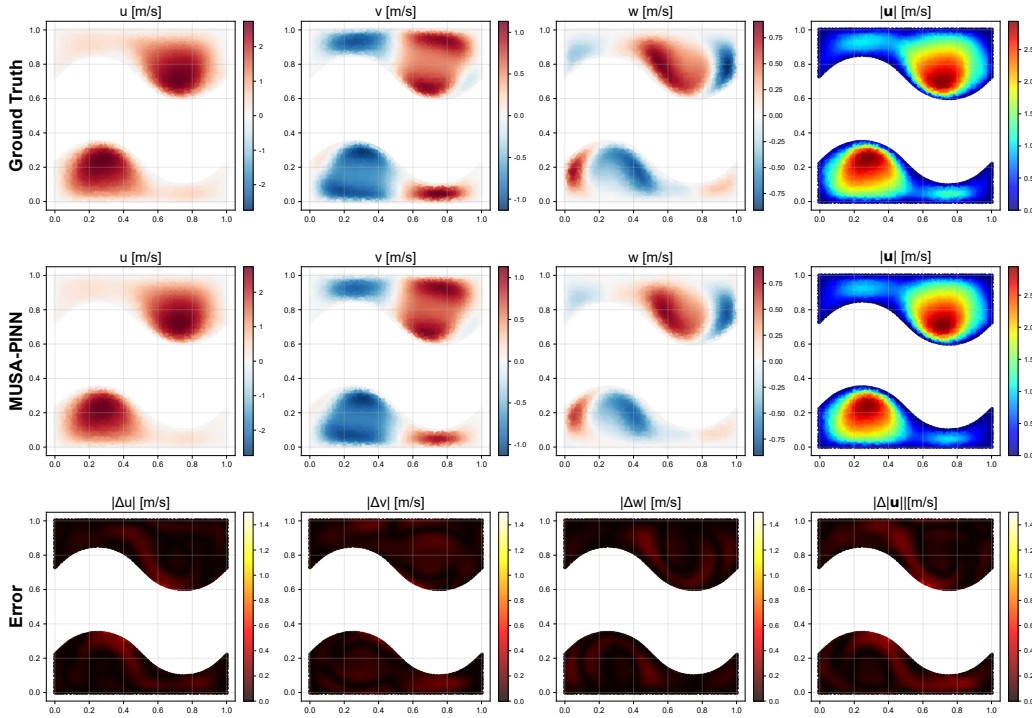

*Figure 15.* **Local flow fidelity at** $x = 0.5$ **(MUSA-PINN).** Component-wise velocity slices $(u, v, w)$ predicted by MUSA-PINN compared with the ground truth (GT) on the plane $x = 0.5$.

