# OpenReview forum: "MUSA-PINN: Multi-scale Weak-form Physics-Informed Neural Networks for Fluid Flow in Complex Geometries"
_ICML.cc/2026/Conference — ICML 2026 regular_

### Official Review · Reviewer_Ntkd · 2026-03-01

**Soundness:** 3
**Presentation:** 3
**Significance:** 3
**Originality:** 3
**Overall Recommendation:** 4
**Confidence:** 5

**Summary:**

This paper proposes MUSA-PINN, a hybrid strong–weak form physics-informed neural network that enforces mass and momentum conservation via surface flux integrals over hierarchically placed spherical control volumes, targeting steady incompressible flow in topologically complex 3D geometries (TPMS channels).

**Compliance With Llm Reviewing Policy:**

Affirmed.

**Key Questions For Authors:**

W1. The paper frames its weak-form formulation as a primary contribution, but applying the divergence theorem to convert strong-form PDE residuals into surface flux integrals over control volumes is the foundational idea behind the finite volume method, which predates PINNs by decades. The V-PINN / hp-VPINN line of work (Kharazmi et al., 2019; 2021) already introduced variational (weak-form) PINNs with domain decomposition. The paper's actual delta over hp-VPINN is (a) using spheres instead of mesh elements and (b) skeleton-guided placement. The paper does not provide any theoretical analysis — no convergence guarantees, no approximation bounds, no analysis of how Monte Carlo quadrature error on the surface integrals propagates to solution error.

W2. All primary experiments are at Re=100 (laminar), steady-state, incompressible, with a single network architecture. The "higher-Re" extension (Section 4.5) only reaches Re=400, still firmly laminar, and reports only two numbers without any baseline comparison.

W3. The paper does not include hp-VPINN (Kharazmi et al., 2021) as a baseline despite it being the most directly comparable weak-form PINN method.

W4. Table 2 ablates scale placement and training schedule exclusively on Diamond. The critical question — whether the skeleton-aware medium-scale placement is actually necessary or whether random placement at the same scale suffices — is never tested. Replacing skeleton centers with random interior centers of the same radius would directly isolate the contribution of topology-awareness, yet this ablation is absent. Additionally, no sensitivity analysis is provided for the numerous hyperparameters: the 12+ loss weights (Table 5) span three orders of magnitude (from 0.1 to 100), the subdomain counts (40/200/500), the radii, and the stage transition epoch at 7000/9000.

**Limitations:**

W1. The paper frames its weak-form formulation as a primary contribution, but applying the divergence theorem to convert strong-form PDE residuals into surface flux integrals over control volumes is the foundational idea behind the finite volume method, which predates PINNs by decades. The V-PINN / hp-VPINN line of work (Kharazmi et al., 2019; 2021) already introduced variational (weak-form) PINNs with domain decomposition. The paper's actual delta over hp-VPINN is (a) using spheres instead of mesh elements and (b) skeleton-guided placement. The paper does not provide any theoretical analysis — no convergence guarantees, no approximation bounds, no analysis of how Monte Carlo quadrature error on the surface integrals propagates to solution error.

W2. All primary experiments are at Re=100 (laminar), steady-state, incompressible, with a single network architecture. The "higher-Re" extension (Section 4.5) only reaches Re=400, still firmly laminar, and reports only two numbers without any baseline comparison.

W3. The paper does not include hp-VPINN (Kharazmi et al., 2021) as a baseline despite it being the most directly comparable weak-form PINN method.

W4. Table 2 ablates scale placement and training schedule exclusively on Diamond. The critical question — whether the skeleton-aware medium-scale placement is actually necessary or whether random placement at the same scale suffices — is never tested. Replacing skeleton centers with random interior centers of the same radius would directly isolate the contribution of topology-awareness, yet this ablation is absent. Additionally, no sensitivity analysis is provided for the numerous hyperparameters: the 12+ loss weights (Table 5) span three orders of magnitude (from 0.1 to 100), the subdomain counts (40/200/500), the radii, and the stage transition epoch at 7000/9000.

**Strengths And Weaknesses:**

S1. The core idea of converting volumetric PDE residuals into surface flux-balance constraints via the divergence theorem, combined with topology-aware (skeleton-guided) control volume placement, is physically well-motivated. The skeleton-aware medium-scale placement in particular is a genuinely useful engineering insight that connects geometric topology to constraint propagation — this is more principled than random subdomain placement.

S2. The empirical gains on the harder geometries (Diamond, IWP) are substantial, and the mass conservation analysis in Figure 5 provides compelling evidence that integral constraints address a real failure mode of pointwise PINNs. The streamline visualizations (Figures 1, 7) effectively communicate the practical impact.

W1. The paper frames its weak-form formulation as a primary contribution, but applying the divergence theorem to convert strong-form PDE residuals into surface flux integrals over control volumes is the foundational idea behind the finite volume method, which predates PINNs by decades. The V-PINN / hp-VPINN line of work (Kharazmi et al., 2019; 2021) already introduced variational (weak-form) PINNs with domain decomposition. The paper's actual delta over hp-VPINN is (a) using spheres instead of mesh elements and (b) skeleton-guided placement. The paper does not provide any theoretical analysis — no convergence guarantees, no approximation bounds, no analysis of how Monte Carlo quadrature error on the surface integrals propagates to solution error.

W2. All primary experiments are at Re=100 (laminar), steady-state, incompressible, with a single network architecture. The "higher-Re" extension (Section 4.5) only reaches Re=400, still firmly laminar, and reports only two numbers without any baseline comparison.

W3. The paper does not include hp-VPINN (Kharazmi et al., 2021) as a baseline despite it being the most directly comparable weak-form PINN method.

W4. Table 2 ablates scale placement and training schedule exclusively on Diamond. The critical question — whether the skeleton-aware medium-scale placement is actually necessary or whether random placement at the same scale suffices — is never tested. Replacing skeleton centers with random interior centers of the same radius would directly isolate the contribution of topology-awareness, yet this ablation is absent. Additionally, no sensitivity analysis is provided for the numerous hyperparameters: the 12+ loss weights (Table 5) span three orders of magnitude (from 0.1 to 100), the subdomain counts (40/200/500), the radii, and the stage transition epoch at 7000/9000.

---

> ### Author Rebuttal · Authors · 2026-03-31
>
> Thank you for your thoughtful review and insightful questions. They have significantly improved this work. We respond to them below:
>
> **Q1: What is the real novelty beyond prior variational PINNs ideas?**
>
> A1: We agree that weak-form supervision is not new. Our contribution is **not** simply using spheres or skeleton-guided placement, but making weak-form conservation **effective** in long, tortuous 3D channels through a **multi-scale strong–weak design**, **topology-aware placement**, and a **two-stage schedule**. Table 2 shows that **single-scale** weak-form variants and **joint training without staging** perform much worse than the full method. Thus, the gain comes from **how** weak-form conservation is deployed, not from conservation alone.
>
> **Q2: Can the paper provide stronger theoretical support?**
>
> A2: We agree that the current paper does not establish a formal convergence theorem, and we will state this explicitly in the revision. Our theoretical support is instead **consistency-based**, following the **Q2/Q3 viewpoint of De Ryck et al. [1]**: Q2 asks whether small residuals imply small solution error, and Q3 asks whether small training loss together with sufficiently many samples implies small generalization error. In our setting, the weak losses in MUSA-PINN are derived from the same steady incompressible Navier--Stokes equations via control-volume integration and the divergence theorem, so the exact solution has zero weak residual on every admissible control volume. The Monte Carlo surface quadrature and component-area estimators used to evaluate these losses are consistent under area-uniform sampling. We will revise the paper accordingly and leave a full residual-to-solution and Monte Carlo error-propagation analysis to future work.
>
> [1] De Ryck et al. Error estimates for physics-informed neural networks approximating the Navier–Stokes equations. 2024.
>
> **Q3: Can the authors provide baseline comparisons at higher Reynolds number, e.g., Re=400, rather than only reporting MUSA-PINN?**
>
> A3: To address this concern directly, we have added the same four baselines used in the main experiments under Re=400 setting and the same matched training protocol. The results, summarized in Table R2 below, show that MUSA-PINN retains its advantage at Re=400. We will include these results in the revision and update the discussion accordingly.
>
> Table R2. Comparison on Gyroid at Re=400 under the same training protocol.
>
> |Method|Rel. ℓ2 on \|u\||Rel. ℓ2 on p|
> |-|-:|-:|
> |PINN-PE|88.02%|89.37%|
> |RoPINN|76.79%|94.16%|
> |CoPINN|85.26%|98.13%|
> |MDPINN-GD|69.33%|61.51%|
> |MUSA-PINN|21.08%|8.09%|
>
> **Q4: Why is hp-VPINN not included as a baseline?**
>
> A4: Since hp-VPINN is one of the most directly related weak-form PINN baselines, we have now added it under the same setting as Table 1. As shown in Table R3, hp-VPINN performs poorly across all tested geometries. We believe the main reason is that, in these long, tortuous 3D channels, weak-form supervision alone is insufficient; strong-form anchoring remains important for stable optimization. Supporting this, on the simplest Primitive geometry, a variant of our method using only small-scale weak-form constraints and no two-stage schedule also remains poor, with relative $\ell_2$ errors of **97.02\% / 92.75\%** on velocity magnitude / pressure. This supports the same conclusion: pure weak-form training is difficult in our setting. We will include the hp-VPINN comparison and clarify this point in the revision.
>
> Table R3. Comparison with hp-VPINN under the same setting as Table 1.
>
> |Geometry|Rel. ℓ2 on \|u\||Rel. ℓ2 on p|
> |-|-:|-:|
> |Primitive|84.51%|39.13%|
> |Gyroid|94.10%|93.04%|
> |Diamond|94.93%|95.74%|
> |IWP|96.94%|102.50%|
>
> **Q5: Does the current ablation really prove that skeleton-aware placement matters?**
>
> A5: To isolate the role of **skeleton-aware medium-scale placement**, we tested **(i)** removing the medium-scale weak form and **(ii)** replacing skeleton-guided centers with random interior centers of the same radius and count. Both substantially increase the **velocity** error: the relative $\ell_2$ error on $|u|$ changes from **12.46\%** to **19.16\%** and **18.28\%**, respectively. This provides direct evidence that **skeleton-guided medium-scale control volumes are important** and more effective than random placement at the same scale.
>
> **Q6: How sensitive is the method to hyperparameters?**
>
> A6: We use a **fixed parameter scheme** across experiments rather than per-geometry retuning. To assess sensitivity, we conducted a **group-wise perturbation study** by applying multiplicative perturbations around the default setting. As shown in **Table R1 in our response to Reviewer ZoPA, Q2**, different parameter groups have different sensitivity, since our primary goal is accurate flow reconstruction, we prioritize velocity fidelity. For new domains, we recommend first normalizing the geometry to $[0,5]^3$ and then starting from the same parameter order.

---

> > ### Author Rebuttal · Reviewer_Ntkd · 2026-04-05
> >
> > The authors have resolved my questions, and congratulations to the authors on the acceptance of their paper.

---

> > > ### Author Response · Authors · 2026-04-05
> > >
> > > Dear Reviewer Ntkd,
> > >
> > > Thank you very much for your thoughtful review and encouraging feedback.
> > >
> > > We are glad that our rebuttal has adequately addressed your concerns. We will incorporate the corresponding clarifications and additional results into the revised manuscript.

---

### Official Review · Reviewer_ZoPA · 2026-03-10

**Soundness:** 3
**Presentation:** 4
**Significance:** 3
**Originality:** 3
**Overall Recommendation:** 5
**Confidence:** 4

**Summary:**

A novel method, "Multi-scale Weak-form Physics-Informed Neural Networks," is proposed for modeling three-dimensional steady-state flows in channels of complex geometry. The method is distinguished by adding integral components (Mass Conservation, Momentum Balance) at different scales to the loss function and dividing the training into multiple stages. It builds upon the traditional Physics-Informed Neural Networks (PINN) approach. The authors demonstrate a significant error reduction compared to other PINN-based models, shown for four complex flow domains.

**Compliance With Llm Reviewing Policy:**

Affirmed.

**Ethical Review Concerns:**

yes

**Final Justification:**

When evaluating the paper, weaknesses were identified in the comparison of the presented method with other solutions in terms of performance, as well as in the detailed discussion of overcoming existing limitations. In their rebuttal, the authors provided new data and expanded the discussion of limitations, which addressed these shortcomings. I recommend accepting the paper.

**Key Questions For Authors:**

1. Does your method offer advantages compared to CFD? If not, what future directions for your method do you envision in this context?
2. Your method uses a multi-component loss function, where each individual term is weighted. In many PINN-based works, this stage is associated with difficulties in selecting these weights. How were the weights selected in your work? Do you expect the selected weights to perform equally well for new flow domains?
3. What potential solutions do you see for overcoming the limitations you identified?

**Limitations:**

yes

**Strengths And Weaknesses:**

Strengths:

1. A significant advantage of the method is demonstrated on various benchmarks.
2. The importance of individual components of the approach is shown in an Ablation Study.
3. The paper is well-structured and easy to read.

Weaknesses:

1. The computational performance of the method is not presented in the main text.
2. The paper mentions limitations but does not discuss potential ways to address them.

---

> ### Author Rebuttal · Authors · 2026-03-31
>
> Thank you for your positive assessment and for the constructive questions. We are encouraged that the reviewer finds the method technically solid, the ablation study informative, and the paper well written. Below we address the main concerns directly.
>
> **Q1: Does the method offer clear practical advantages over conventional CFD? If not, what role do the authors envision for it in the future?**
>
> A1: We do not position MUSA-PINN as a replacement for mature CFD for every forward solve. Its advantage is different: it is **volumetric-mesh-free**, avoiding expensive body-fitted volumetric meshing in complex 3D channels, and once trained it provides a **differentiable coordinate-to-field representation**, so spatial gradients are directly available by automatic differentiation. We therefore see its main value in **repeated-query settings** such as design exploration, optimization, and inverse problems, rather than single-shot CFD replacement.
>
> **Q2: How should the multi-component loss weights be chosen, and how robust are they across different flow domains?**
>
> A2: We use a **fixed weighting scheme** rather than per-geometry retuning. The default parameters were originally chosen based on exploratory sensitivity experiments during method development. To make this point explicit, we reran a representative group-wise perturbation study under the final training protocol and summarize the results in Table R1. As shown in **Table R1**, different parameter groups have different sensitivity, and some perturbations improve **pressure** error while worsening **velocity** error. Since our main goal is accurate flow / transport reconstruction in long, tortuous channels, we choose the default setting for the **best overall tradeoff**, with priority on **velocity fidelity** while keeping pressure accuracy reasonable. Similar parameter magnitudes also work across both TPMS channels and larger geometries such as the liquid cooling plate. For new domains, we recommend first normalizing the geometry to $[0,5]^3$ and then starting from the same parameter order.
>
> Table R1. Hyperparameter sensitivity across representative domains.
>
> | Weight group | Scale factor | TPMS case (vel / p err) |
> | - | -: | -: |
> | Default | 1x | 12.46% / 26.55% |
> | Boundary Conditions group | 0.1x | 21.38% / 2.70% |
> | | 0.2x | 18.31% / 20.78% |
> | | 5x | 22.07% / 30.45% |
> | | 10x | 30.21% / 52.97% |
> | Strong form mass | 0.1x | 28.16% / 4.01% |
> | | 0.2x | 16.69% / 21.18% |
> | | 5x | 20.73% / 10.32% |
> | | 10x | 24.10% / 3.01% |
> | Strong form momentum | 0.1x | 23.00% / 69.51% |
> | | 0.2x | 20.91% / 15.89% |
> | | 5x | 15.05% / 27.69% |
> | | 10x | 18.77% / 17.95% |
> | Weak form mass | 0.1x | 16.17% / 24.41% |
> | | 0.2x | 15.74% / 26.35% |
> | | 5x | 20.35% / 7.64% |
> | | 10x | 22.54% / 3.69% |
> | Weak form momentum | 0.1x | 16.66% / 24.46% |
> | | 0.2x | 15.74% / 26.35% |
> | | 5x | 20.35% / 7.65% |
> | | 10x | 16.31% / 25.21% |
> | Sphere Radius | 0.67x | 21.20% / 5.73% |
> | | 0.77x | 17.03% / 23.75% |
> | | 1.3x | 17.80% / 22.84% |
> | | 1.5x | 24.21% / 9.99% |
>
> **Q3: What concrete solutions do the authors envision for overcoming the limitations already acknowledged in the paper?**
>
> A3: We agree that the limitations discussion should be more actionable. The current paper identifies two main limitations: **additional training cost** from Monte Carlo integration on multi-scale control-volume surfaces, and **heuristic control-volume placement**. We will revise this part to state concrete solutions more explicitly. Specifically, to reduce training cost, we plan to make the weak-form supervision **adaptive and sparse**, e.g., by activating only the most informative control volumes based on residual, rather than evaluating all of them uniformly throughout training. To address the heuristic placement issue, we plan to learn control-volume placement from geometry and training signals instead of fixing it heuristically.
>
> **Q4: Why is the computational performance not presented more explicitly in the main text?**
>
> A4: We do report computational performance in **Appendix D.3 / Table 6**, including convergence iterations and wall-clock training time for the baselines and the two stages of MUSA-PINN. We will move a concise summary of this **accuracy–efficiency tradeoff** into the main text. MUSA-PINN incurs extra training cost because weak-form supervision requires **Monte Carlo integration on multi-scale control-volume surfaces** and a **two-stage schedule**, but this overhead yields substantially better stability and accuracy on the most challenging geometries.

---

> > ### Author Rebuttal · Reviewer_ZoPA · 2026-04-03
> >
> > The author’s responses fully address the questions raised in the review.

---

> > > ### Author Response · Authors · 2026-04-04
> > >
> > > Dear Reviewer ZoPA,
> > >
> > > Thank you again for your constructive suggestions and positive feedback.
> > >
> > > We will incorporate the corresponding clarifications and additional results into the revised manuscript.

---

### Official Review · Reviewer_YYe7 · 2026-03-10

**Soundness:** 3
**Presentation:** 3
**Significance:** 2
**Originality:** 2
**Overall Recommendation:** 4
**Confidence:** 5

**Summary:**

This paper addresses fluid flow in complex geometries using a PINN residual based approach. It extends the PINN by using weak form constraints over spherical control volumes. PINNS have traditionally been applied to quite simple problems, and this paper tries to extend them to 'industrial-grade' applications with complex geometries. It claims that enforcing constraints leads to better, and more robust, PINN solvers.  This method is tried out in some simple geometries.

**Compliance With Llm Reviewing Policy:**

Affirmed.

**Final Justification:**

I'm  happy for the paper to be accepted

**Key Questions For Authors:**

1. Conservation laws plus PINNs have been very well studied by many authors and are at the heart of the DRM method and also in the workings  of physics informed neural operators. How does this paper differ from these?

2. Are the geometries considred really industrial geometries. Sure someting like a gas turbine would be more realistic

3. How does the method compare with FE or FV in similar geometries?

4. Can the authors give any sort of convergence analysis?

**Limitations:**

1. As is often the case the method proposed is claimed to be mesh free. In practice all PINNS rely on a mesh either directly through the points where residuals are calculated or indirectly through the constraints imposed by the PINN architecture.

2. The convergence theory for PINNS is currently weak to non-existent. How will this be addressed by this methodology.

3. There is no comparison with FE or similar methods

**Strengths And Weaknesses:**

This method extends the standard PINN approach by building in conservation laws across some intersting (but not especially complex) geometries. The results are good on the ablation studies

The methodology is fairly standard and the loss is similar to many other PINN type applications. The constrint is built into the loss. This is also a very standard approach which has many problems associated with it. The flows considered are all fairly simple

No comparison is made with FE or similar mathods which wold be easy to implement in these geometries

---

> ### Author Rebuttal · Authors · 2026-03-31
>
> We greatly appreciate your thoughtful review and insightful questions. Below is our point-by-point response.
>
> **Q1: Conservation laws plus PINNs have been studied by many authors and are at the heart of the DRM method and also in the workings of physics informed neural operators. How does this paper differ from these?**
>
> A1: We agree that conservation-law supervision is not new. DRM optimizes a global variational/Ritz objective, whereas MUSA-PINN is a hybrid strong–weak PINN that retains the strong-form Navier–Stokes residuals and adds local control-volume flux-balance constraints. Our point is that **conservation alone is not enough**: Table 2 shows that single-scale or naively imposed conservation performs much worse than our full method. The improvement comes from combining conservation with **multi-scale weak-form control volumes, topology-aware placement, and two-stage schedule**, which together make conservation effective in long, tortuous complex geometry channels.
>
> **Q2: Are these really industrial geometries?**
>
> A2: We agree that our wording should be more precise. Rather than simplified benchmarks, the TPMS geometries represent practical and widely studied configurations for **compact heat exchangers** and **thermal-management devices** [1]. Specifically, recent work studies **compact heat exchangers with internal TPMS geometry** [2], and TPMS heat exchangers have also been investigated for **micro gas turbines** [3]. Our claim is therefore not that we already cover all industrial components such as full gas-turbine hardware, but that we address an **industrially relevant internal-flow setting** that current PINN methods struggle with. These long, tortuous, interconnected 3D passages are exactly the type of geometry that motivates our method. We will revise the wording.
>
> [1] Amara et al. Review of Triply Periodic Minimal Surface (TPMS) Structures for Cooling Heat Sinks. 2025.
>
> [2] Montenegro et al. Modelling of a Compact Heat Exchanger With TPMS Geometry. 2025.
>
> [3] Su et al. Experimental and numerical study on flow and heat transfer characteristics of additively manufactured triply periodic minimal surface (TPMS) heat exchangers for micro gas turbine. 2025.
>
>
> **Q3: How does the method compare with FE or FV in similar geometries?**
>
> A3: The reference solutions in this paper are already generated by **Ansys CFX**, i.e., a mature commercial CFD solver based on the **FVM**. For steady 3D Navier–Stokes flow in such complex geometries, closed-form analytical solutions are generally unavailable, so high-fidelity CFD is the standard source of ground truth. Our goal here is therefore not to replace mature FE/FV solvers, but to show that PINNs in this regime can be made substantially more stable and physically consistent. Compared with conventional CFD workflows, MUSA-PINN is **volumetric-mesh-free** and, once trained, provides a **differentiable coordinate-to-field surrogate**, so spatial gradients can be obtained directly by automatic differentiation.
>
> **Q4: Can the authors give any sort of convergence analysis?**
>
> A4: We agree that the current paper does not prove a formal convergence theorem, and we will make this explicit in the revision. Our theoretical support is instead **consistency-based**, in the **Q2/Q3 viewpoint of De Ryck et al. [4].**: Q2 asks whether small residuals imply small solution error, and Q3 asks whether small training loss together with sufficiently many samples implies small generalization error. In our setting, MUSA-PINN is a **hybrid strong–weak** formulation, where the weak continuity and momentum losses are obtained by integrating the same steady incompressible Navier–Stokes equations over clipped control volumes and applying the divergence theorem. Hence, the exact solution makes both the pointwise strong residuals and the control-volume weak residuals vanish. The weak losses are evaluated by Monte Carlo surface quadrature, and the associated component-area estimators are consistent under area-uniform sampling. We will revise the paper accordingly and leave a full residual-to-solution and Monte Carlo error-propagation analysis to future work.
>
> [4] De Ryck et al. Error estimates for physics-informed neural networks approximating the Navier–Stokes equations. 2024.
>
> **Q5: In what sense is the method mesh-free in practice?**
>
> A5: We agree that “mesh-free” should be stated precisely. Our claim is not that MUSA-PINN avoids all discretization or sampling. Rather, it is **volumetric-mesh-free**: it does **not** require a body-fitted volumetric mesh, element connectivity, or element-based quadrature, which are typically  time-consuming and expertise-intensive parts of CFD pipelines for slender, tortuous 3D channels. Instead, we use point sampling for strong-form supervision and Monte Carlo sampling on control-volume boundaries for weak-form supervision. The geometry is represented by a **watertight boundary triangle mesh**, which is used to support clipped-boundary sampling.

---

> > ### Author Rebuttal · Reviewer_YYe7 · 2026-04-01
> >
> > I am happy that the authors have addressed my concerns. In view of this, and also taking into account the responses to the other reviewers I will raise my score.

---

> > > ### Author Response · Authors · 2026-04-04
> > >
> > > Dear Reviewer YYe7,
> > >
> > > We sincerely thank you for your encouraging feedback and for raising your score.
> > >
> > > We will incorporate the corresponding clarifications and further elaborations into the revised manuscript.

---

### Official Review · Reviewer_koRR · 2026-03-13

**Soundness:** 2
**Presentation:** 4
**Significance:** 3
**Originality:** 2
**Overall Recommendation:** 5
**Confidence:** 4

**Summary:**

MUSA-PINN is a mesh-free physics-informed neural network for solving the steady-state flow for the incompressible Navier-Stokes equation in topologically complex geometries. The key insight is that prior work struggles with complex geometries due to pointwise strong-form residual minimization; the current work adds weak-form integral equations (at multiple scales) to improve convergence. The method is primarily validated across multiple TPMS geometries and compared against recent PINNs, significantly reducing relative errors with reference CFD solutions.

**Compliance With Llm Reviewing Policy:**

Affirmed.

**Final Justification:**

After rebuttal response and reading other reviews, I stand by my original positive score.

**Key Questions For Authors:**

1.	Note that there are other neural methods, like Monte-Carlo methods, for solving Navier-Stokes. A recent example is “Neural Monte Carlo Fluid Simulation” by Jain et al. in SIGGRAPH 2024. Could you please position your work with regard to these, and additionally provide a comparison?
2.	In Table 2, can you add a strong-form baseline? I believe the strong form is either similar to or identical to PINN-PE. But as noted in Table 2, the “Small-only” variant gives a higher error than PE-PINN (as in Table 1). Can you please justify this observation? Moreover, I am curious to see how much this (small-scale subdomain integral) adds, considering you already have a strong form.
3.	What happens if you only keep the weak form? It's possible that the strong form is redundant. Note that there are works that use only weak forms of the PDEs, for example, “NeuralClothSim: Neural Deformation Fields Meet the Thin Shell Theory” by Kairanda et al., NeurIPS 204.
4.	It is not explained why the spatio-temporal fluid flow is excluded from the current work. What is the reason for sticking with steady-state flow? Preliminary results are welcome here.

**Limitations:**

yes

**Strengths And Weaknesses:**

Soundness:

++ All the formulations in the paper appear correct and sound. The ideas of using weak form integral equations for supporting long-range conservation, and multi-scale volume at global/skeleton/local is well-motivated and correctly formulated.

++ The claims are well supported by experimental results where the authors show multiple geometries of varying difficulties and even industrial scale applications.

-- There are a few experiments that would add value, see the questions section.

Presentation:

++ The paper is well-structured, well-written, and easy to read and understand. Figures, tables, and written – all nice!

-- I found the abstract to be a bit misleading, although all other parts seem fine. While the proposed approach works only for fluid flow, the current version of subtract suggests that the method works for all PDEs. I highly recommend rewriting.

-- There are other works explored previously for weak form, but only Kharazmi 2019, 2021 is cited and discussed in related works. Please see surveys like “Physics-Informed Machine Learning: A Survey on Problems, Methods and Applications” by Hat et al.

Significance:

++ The work falls under scientific machine learning that could address the shortcomings of conventional computational fluid dynamics. Thus, significant!

Originality:

++ The method provides new insights – how to overcome the challenges of strong-form residuals that struggle to enforce the PDE pointwise complex geometries. The insight is to use weak-form global formulations. The other interesting takeaway is how to set the multi-scales using geometric heuristics.

---

> ### Author Rebuttal · Authors · 2026-03-31
>
> Thank you for your positive assessment and constructive suggestions. Below are our responses to the questions raised:
>
> **Q1: How does the proposed method relate to other neural methods for fluid simulation, such as Neural Monte Carlo Fluid Simulation?**
>
> A1: We thank the reviewer for pointing out NMC and will add it to the discussion. While both methods leverage neural networks to accelerate fluid simulations, NMC is not a like-for-like baseline for our setting: NMC studies time-dependent simulation based on the **inviscid Euler equation**, with no-penetration velocity, Neumann pressure, and specified initial conditions evolved over time. In contrast, our work targets **steady incompressible viscous Navier–Stokes** in complex internal-flow geometries, with prescribed inlet velocity, outlet pressure, and no-slip walls. We therefore view NMC as related but not directly comparable under the same physical setup.
>
> **Q2: Should Table 2 include a strong-form baseline? Why does the Small-only variant perform worse, and what does the small-scale weak form contribute?**
>
> A2: The strong-form baseline is already included as **PINN-PE** in Table 1. MUSA-PINN starts from this strong-form solver and adds weak-form conservation losses while retaining the strong-form residuals in the hybrid objective. We will clarify this in the revision and, space permitting, add the PINN-PE reference near Table 2.
>
> **Small-only** performs poorly because small control volumes mainly provide **local refinement**, whereas the main challenge in these geometries is **long-range conservation propagation** through tortuous channels. In our design, large-scale subdomains enforce global coupling, medium-scale subdomains cover transport pathways via the skeleton, and small-scale subdomains refine local details.
>
> We further added an ablation that removes the **small-scale control volumes** while keeping the larger and medium scales. On the Diamond case, performance degrades from **12.46\% / 26.55\%** to **16.72\% / 24.44\%** on velocity magnitude / pressure, showing that the small-scale weak form is **necessary** for local refinement and improves the full multi-scale solution.
>
> **Q3: What happens if only the weak form is kept?**
>
> A3: Our method is intentionally a **hybrid strong–weak** formulation. The strong-form residual provides **pointwise** physics supervision and serves as the baseline constraint in the objective, while the weak-form terms are derived from the same Navier–Stokes equations and are added as **complementary supervision**, rather than replacing the strong form.
>
> To directly answer the reviewer’s question, we additionally ran a **pure weak-form** ablation by removing the strong-form loss $\mathcal{L}_{\mathrm{sf}}$ and keeping only boundary supervision and weak-form constraints. On the Diamond case, this variant performs worse than the full hybrid model: the relative $\ell_2$ error changes from **12.46\% /  26.55\%** to **43.92\% / 93.81\%** on velocity magnitude / pressure. This confirms that the strong form is **not redundant** in our setting. This is also consistent with the poor performance of hp-VPINN reported in Table R3 of our response to Reviewer Ntkd, Q4, further suggesting that weak-form supervision alone is insufficient in long, tortuous 3D channels.
>
> **Q4: Why does the paper focus on steady-state flow rather than spatiotemporal flow?**
>
> A4: We focus on **steady incompressible internal flow** because this is already a practically important regime for industrial component design, including heat exchangers and cooling plates, where optimization is often driven by **steady-state thermo-hydraulic metrics** such as pressure drop, temperature distribution, and efficiency[1]. For example, recent heat-exchanger studies evaluate TPMS heat exchangers with steady-state experimental and numerical analysis[2][3].
>
> [1] Feng et al. Triply periodic minimal surface (TPMS) porous structures: from multi-scale design, precise additive manufacturing to multidisciplinary applications. 2022.
>
> [2] Yan et al. Numerical investigation into thermo-hydraulic characteristics and mixing performance of triply periodic minimal surface-structured heat exchangers. 2023.
>
> [3] Wang et al. Investigation on flow and heat transfer in various channels based on triply periodic minimal surfaces (TPMS). 2023.
>
> **Q5: The paper’s scope and positioning could be stated more precisely: the abstract may sound broader than the actual setting, and the related-work discussion on weak-form methods could be more complete.**
>
> A5: We agree. This paper specifically studies **steady incompressible Navier–Stokes flow in complex 3D internal-flow geometries**, and we will revise the abstract and introduction to make this scope explicit and avoid over-generalization.
>
> We will also expand the related-work discussion by adding more weak-form and survey references, while clarifying our distinction from prior strong-form, weak-form, and multi-scale PINN methods.

---

> > ### Author Rebuttal · Reviewer_koRR · 2026-04-03
> >
> > Thanks for the rebuttal. All my concerns are adequately addressed except for one: "Why does the small-only variant (which includes strong form & small-only weak form) perform worse than the strong-form version?" (cf. Tables. 1, 2)

---

> > > ### Author Response · Authors · 2026-04-04
> > >
> > > Dear Reviewer koRR,
> > >
> > > Thank you very much for the follow-up and for engaging deeply with our work!
> > >
> > > Our interpretation is that the issue is not weak-form supervision itself, but adding **only small-scale** weak-form constraints to a strong-form PINN that is already dominated by local objectives. In this case, the added small-scale weak-form loss remains **highly local** and biases training toward satisfying local surface-flux constraints, without restoring the **long-range transport consistency** missing from the strong-form baseline. As a result, the added small-scale weak-form loss can steer the model toward locally consistent but globally mismatched solutions. In our ablation, this mismatch is reflected most clearly in pressure, while the velocity error remains comparable to the strong-form baseline.
> > >
> > > The full model avoids this failure mode because the larger and skeleton-aware control volumes provide the nonlocal coupling absent in the small-only variant.

---

### Decision · Program_Chairs · 2026-04-30

**Decision:**

Accept (regular)

**Comment:**

This paper introduces MUSA-PINN, a hybrid strong–weak PINN with multi-scale control-volume constraints for complex geometries. Reviewers agree the method is technically sound, well presented, and shows strong empirical improvements on challenging cases.

Concerns focused on limited novelty relative to prior weak-form PINNs and missing evaluations (e.g., hp-VPINN, higher-Re settings, and key ablations). The authors addressed these points convincingly in the rebuttal with additional experiments and clearer positioning.

Overall, despite somewhat incremental novelty, the paper provides a solid and practically useful advance for PINNs in complex domains.